# Dopamine activity encodes the changing valence of the same stimulus in conditioned taste aversion paradigms

Maxine K Loh[1,2†], Samantha J Hurh[2†], Paula Bazzino[3], Rachel M Donka[2], Alexandra T Keinath[2], Jamie D Roitman[2,3], Mitchell F Roitman[2,3*]

[1]Division of Endocrinology, Diabetes and Metabolism, Department of Medicine, University of Illinois at Chicago, Chicago, United States; [2]Department of Psychology, University of Illinois at Chicago, Chicago, United States; [3]Graduate Program in Neuroscience, University of Illinois at Chicago, Chicago, United States

## eLife Assessment

This study utilizes an elegant approach to examine valence encoding of the mesolimbic dopamine system. The findings are **valuable**, demonstrating differential responses of dopamine to the same taste stimulus according to its valence (i.e., appetitive or aversive) and in alignment with distinct behavioral responses. The evidence supporting the claims is **convincing**, resulting from a well-controlled experimental design with minimal confounds and thorough reporting of the data.

*For correspondence:
mroitman@uic.edu

†These authors contributed equally to this work

Competing interest: The authors declare that no competing interests exist.

**Abstract** Mesolimbic dopamine encoding of non-contingent rewards and reward-predictive cues has been well established. Considerable debate remains over how mesolimbic dopamine responds to aversion and in the context of aversive conditioning. Inconsistencies may arise from the use of aversive stimuli that are transduced along different neural paths relative to reward or the conflation of responses to avoidance and aversion. Here, we made intraoral infusions of sucrose and measured how dopamine and behavioral responses varied to the changing valence of sucrose. Pairing intraoral sucrose with malaise via injection of lithium chloride (LiCl) caused the development of a conditioned taste aversion (CTA), which rendered the typically rewarding taste of sucrose aversive upon subsequent re-exposure. Following CTA formation, intraoral sucrose suppressed the activity of ventral tegmental area dopamine neurons ($VTA_{DA}$) and nucleus accumbens (NAc) dopamine release. This pattern of dopamine signaling after CTA is similar to intraoral infusions of innately aversive quinine and contrasts with responses to sucrose when it was novel or not paired with LiCl. Dopamine responses were negatively correlated with behavioral reactivity to intraoral sucrose and predicted home cage sucrose preference. Further, dopamine responses scaled with the strength of the CTA, which was increased by repeated LiCl pairings and weakened through extinction. Thus, the findings demonstrate differential dopamine encoding of the same taste stimulus according to its valence, which is aligned to distinct behavioral responses.

## Introduction

Dopamine neurons of the ventral tegmental area ($VTA_{DA}$) and their release of dopamine in the nucleus accumbens (NAc) play key roles in the encoding of primary reward (*Cohen et al., 2012*; *Roitman et al., 2008*; *Romo and Schultz, 1989*), reward-related learning (*Schultz, 2016*; *Schultz et al., 1997*; *Watabe-Uchida et al., 2017*), and motivation (*Berridge and Robinson, 1998*; *Bromberg-Martin et al., 2010*; *Wise, 2004*). Brief, phasic dopamine responses are evoked by primary rewarding stimuli

and develop to their predictors (*Konanur et al., 2024*, for example). Pauses in dopamine activity are evoked by reward omission (*Sugam et al., 2012*; *Tobler et al., 2003*). Collectively, these findings support a role for dopamine in signaling reward prediction errors (*Schultz, 2016*; *Schultz, 1998*; *Schultz et al., 1997*; *Steinberg et al., 2013*). This value-learning account has been more recently challenged to include the learning from, and updating of, the value of reward-directed actions (*Coddington et al., 2023*; *Jeong et al., 2022*). Yet there is little debate that dopamine increases to reward and reward-related cues. A role for dopamine in aversion and aversive conditioning is more controversial (*Morales and Margolis, 2017*). Some recordings of $VTA_{DA}$ activity (*Bromberg-Martin et al., 2010*; *Matsumoto and Hikosaka, 2009*) or dopamine release (*Mikhailova et al., 2019*; *Wenzel et al., 2015*) support increased signaling in aversion and aversive conditioning while others support decreased signaling (*Mileykovskiy and Morales, 2011*; *Roitman et al., 2008*; *Ungless et al., 2004*; *Wheeler et al., 2011*; *Zhuo et al., 2024*; see *Morales and Margolis, 2017* for review).

The unsettled nature of dopamine signaling in aversion could be due to several factors. First, like reward (*Berridge and Robinson, 2003*), aversion is a multi-dimensional construct that ranges from involuntary responses to noxious primary stimuli to passive or active avoidance in response to predictors of aversion. Aversive stimuli have included foot/tail shocks (*de Jong et al., 2019*; *Mileykovskiy and Morales, 2011*), air puffs (*Matsumoto and Hikosaka, 2009*; *Mirenowicz and Schultz, 1996*; *Zhuo et al., 2024*), painful pinches (*Romo and Schultz, 1989*), and white noise (*Goedhoop et al., 2022*). These are transduced along different sensory pathways (i.e., pain, somatosensory, auditory) from reward (typically gustatory; e.g., sucrose). Even when administering rewarding and aversive stimuli that utilize the same sensory pathway, as with taste, rewarding sweet and aversive bitter solutions activate different taste receptors (*Schier and Spector, 2019*). Complicating the picture further, meso-limbic dopamine responses to noxious stimuli have been assayed in anesthetized animals (*Brischoux et al., 2009*; *Budygin et al., 2012*; *Romo and Schultz, 1989*) – making behavioral assessment of aversion impossible. In awake, behaving recordings, behavioral paradigms often permit the avoidance of aversive stimuli (*Kutlu et al., 2021*; *Oleson et al., 2012*), which can ultimately be rewarding. To facilitate the sampling of aversive stimuli, water or food deprivation protocols are often used (*Glover et al., 2016*; *Gordon-Fennell et al., 2023*; *Hurley et al., 2023*; *López et al., 2023*; *Miranda et al., 2023*). However, dopamine responses are modulated by physiological state (i.e., hunger, thirst, sodium appetite *Cone et al., 2015*; *Fortin and Roitman, 2018*; *Hsu et al., 2020*), presenting a further confound in interpreting mesolimbic responses.

To overcome these limitations, we measured lateral $VTA_{DA}$ neural activity and dopamine release in the lateral shell of the NAc using fiber photometry as we varied the valence of intraoral sucrose infusions. NAc lateral shell dopamine differentially encodes cues predictive of rewarding (i.e., sipper spout with sucrose) and aversive stimuli (i.e., footshock), which is distinct from other subregions (*de Jong et al., 2019*). It is important to note that other regions of the NAc may serve as hedonic hotspots for example, dorsomedial shell; or may more closely align with the signaling of salience (e.g., ventro-medial shell; *Yuan et al., 2019*). In Paired rats, the valence of sucrose was changed by subsequently administering lithium chloride (LiCl) – which induces visceral malaise (*Bernstein et al., 1992*) and reliably conditions a taste aversion (CTA; Garcia et al., 1 955; *Garcia and Kimeldorf, 1957*; *Nachman and Ashe, 1973*; *Nolan et al., 1997*; *Kim et al., 2010*; *Swank and Bernstein, 1994*; *Thiele et al., 1996*). In Unpaired rats, sucrose valence was unchanged by injecting LiCl 24 hr after intraoral sucrose (*Smith and Roll, 1967*). Changes in valence were strengthened through multiple pairings of sucrose and LiCl or weakened through extinction. Throughout, rats were fed and watered ad libitum to avoid physiological need as a confound. To assay the stereotypical appetitive and aversive behavioral responses to taste stimuli (*Breslin et al., 1992*; *Grill and Norgren, 1978a*), we used a deep-learning algorithm (DeepLabCut, *Mathis et al., 2018*) and home cage sucrose preference task to assess the valence of intraoral sucrose. Across testing, $VTA_{DA}$ activity and dopamine release in the NAc differentially responded to sucrose taste based on its affective value ascertained from behavioral reactivity and sucrose preference. Thus, we conclude that the mesolimbic dopamine system differentially encodes valence – reward versus aversion – and flexibly for the same stimulus.

## Results

### NAc dopamine differentially responds to primary taste stimuli and correlates with differential behavioral reactivity

We expressed the dopamine fluorescent sensor AAV1.Syn.Flex.GRAB_DA2h (GRAB_DA2h) in the lateral shell of the NAc and recorded real-time fluorescence via an indwelling fiber optic (*Figure 1A, B*, *Figure 1—figure supplement 1*). To characterize dopamine and behavioral responses to innately appetitive and aversive taste stimuli, we made intraoral infusions of sucrose and quinine in naive rats. On average, intraoral infusions of sucrose and quinine evoked different dopamine release responses ($t(5) = 3.61$, $p < 0.05$; *Figure 1C*), which is consistent with prior work from our group (*Hsu et al., 2020*; *Roitman et al., 2008*). Across individual trials, responses to sucrose infusions were higher than those to quinine (*Figure 1D*). Using receiver operating characteristic (ROC) analysis (*Cone et al., 2015*; *Green and Swets, 1974*), we found dopamine responses to sucrose and quinine delivery on individual trials to be highly discriminable (area under the ROC curve [$AUC_{ROC}$] = 0.81; *Figure 1E*).

Appetitive and aversive tastes arouse well-characterized, stereotypical responses in rats (*Grill and Norgren, 1978a*). Specifically, intraoral infusions of appetitive taste stimuli evoke mouth movements and tongue protrusions, but the head and body are relatively still. In contrast, aversive taste stimuli cause gapes, headshakes, forelimb flails, and chin rubs (*Grill and Norgren, 1978a*) – and, therefore, greater head and whole-body movement. We hypothesized that these responses could be well captured by measuring movement of the nose and forepaws (*Figure 2A*). Here, primary taste stimuli produced distinct nose and forepaw responses to intraoral infusions relative to a 5-second baseline period just before infusion onset (behavioral reactivity, *Figure 2B*). Quinine produced greater average nose ($t(5) = 3.69$, $p < 0.05$, *Figure 2C*); and forepaw ($t(5) = 3.82$, $p < 0.05$; *Figure 2D*) movement relative to sucrose in the same rats. Importantly, these behavioral measures were negatively correlated to dopamine responses from the same rats where greater dopamine responses were associated with less movement evoked by the stimulus (nose movement to dopamine: $r^2 = 0.44$, slope = −0.041; $p < 0.05$; forepaw moment to dopamine: $r^2 = 0.44$, slope = −0.029; $p < 0.05$; *Figure 2E, F*).

### Dopamine responses differentially encode the same taste stimulus based on behavioral reactivity

CTA robustly shifts behavior to the same taste stimulus from ingestion to aversion (*Grill and Norgren, 1978b*). Here, we measured either dopamine release or $VTA_{DA}$ activity (*Figure 3—figure supplement 1*) and behavioral responses from initially naive rats (Conditioning Day, CD) to intraoral infusions of 0.3 M sucrose. We then administered either Saline (Unpaired) or malaise-inducing LiCl (Paired). The subsequent day, rats did not receive intraoral infusions but did receive the counterbalanced injection in their home cage and were untreated the following day (*Figure 3A*). All rats then received another session of sucrose intraoral infusions (Test Day, TD). Thus, by TD, all rats had equal exposure to all stimuli with the lone difference being that Paired rats had sucrose and LiCl administered in close temporal proximity whereas Unpaired rats had sucrose and LiCl administered at least 24 hr apart. On TD, we measured subjects' dopamine and behavioral responses to intraoral sucrose as per CD (*Figure 3A*). In Unpaired rats, dopamine release to intraoral sucrose was unchanged between days (Unpaired: $t(9) = 0.83$, $p > 0.05$; *Figure 3B*). In sharp contrast, in Paired rats NAc dopamine release evoked by intraoral sucrose was significantly suppressed on TD relative to CD (Paired: $t(10) = 3.97$, $p < 0.005$; *Figure 3B*). Using an ROC to test for between-subject differences, we found that dopamine release to intraoral sucrose on individual trials from Paired and Unpaired rats could not be discriminated on CD ($AUC_{ROC} = 0.50$) but could be well discriminated on TD ($AUC_{ROC} = 0.80$; *Figure 3C–F*).

Similar to dopamine release in the NAc, $VTA_{DA}$ activity was unchanged between days in Unpaired rats (Unpaired: $t(5) = 0.60$, $p > 0.05$; *Figure 3G*) but suppressed on TD relative to CD in Paired rats (Paired: $t(6) = 2.627$, $p < 0.05$; *Figure 3G*). Likewise, individual $VTA_{DA}$ responses between groups could not be discriminated on CD ($AUC_{ROC} = 0.55$) but were well discriminated on TD ($AUC_{ROC} = 0.76$; *Figure 3H–K*).

We analyzed behavioral reactivity to sucrose infusions on CD and TD in Unpaired and Paired rats, during which dopamine release or $VTA_{DA}$ activity was also measured. For rats in which NAc dopamine release was measured, intraoral infusions of sucrose elicited different nose movement responses across testing days (Treatment [Unpaired vs. Paired] × Day [CD vs. TD]: $F_{(1,18)} = 6.61$, $p < 0.05$; *Figure 4A*). In

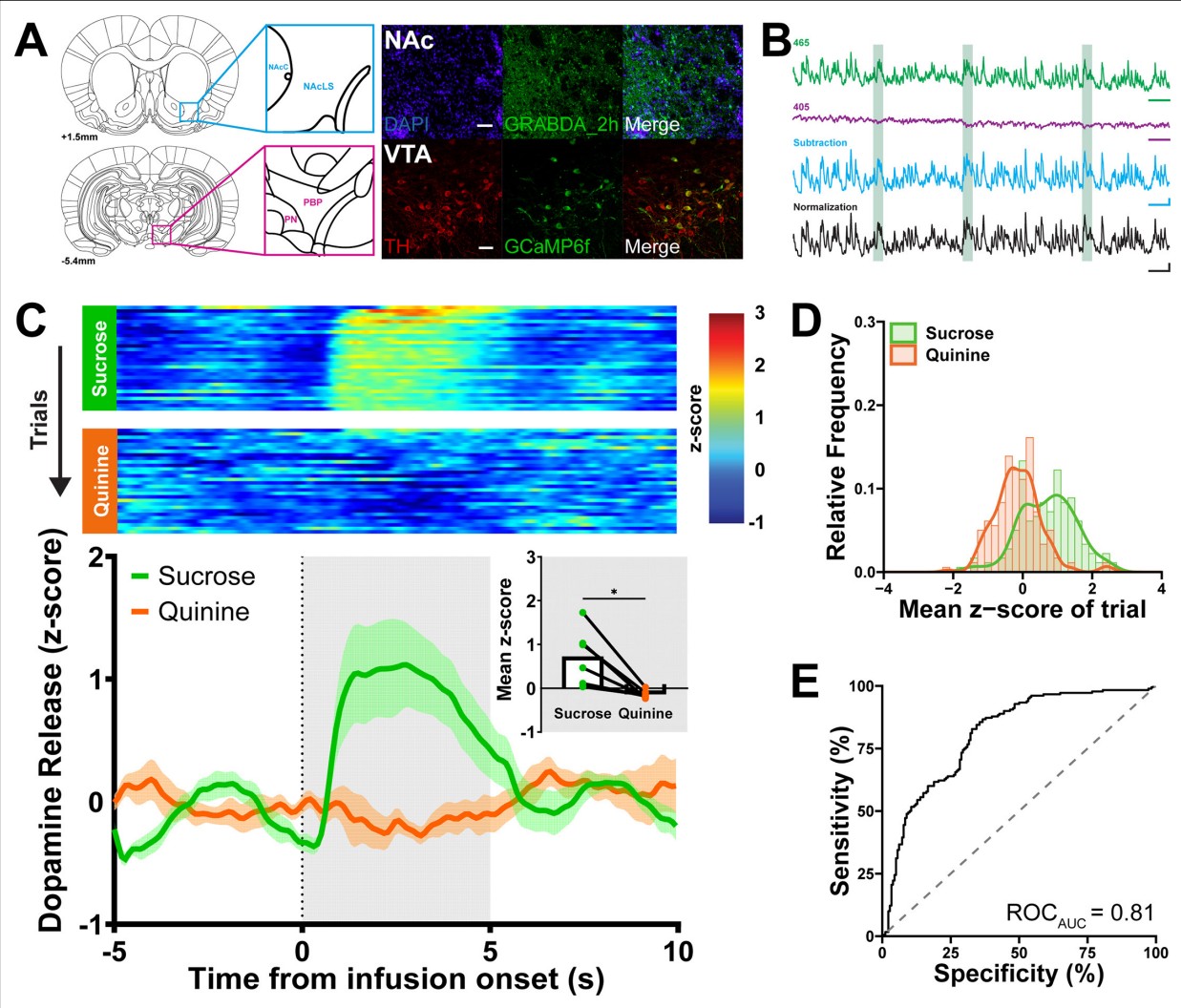

**Figure 1.** In vivo fiber photometry in the mesolimbic dopamine system captures phasic dopamine responses to primary taste stimuli. (**A**) Representative images of GRABDA_2h and Cre-dependent GCamp6f expression in the NAc and VTA, respectively. *Top row*: Dopamine release recordings from the lateral subregion of the NAc shell are confirmed via viral expression of dopamine-sensor, GRABDA2h (green), probed against DAPI (blue) to visualize sensor location targeted to the lateral shell of the NAc (NAcLS), which borders the NAc core (NAcC). *Bottom row*: Dopamine cellular activity was recorded in TH Cre+ rats in the VTA (paranigral nucleus [PN], the parabrachial pigmented area [PBP]). TH+ (red) colocalized with intracellular calcium-sensor, GCaMP6f (green) to demonstrate isolation of the $VTA_{DA}$ population. (**B**) Real-time dopamine release in the NAc across processing steps from a representative rat receiving 5 s 0.3 M sucrose intraoral infusions (light green bars). Fluorescence excited by the 465 nm ($Ca^{2+}$- and GRABDA2h-dependent, green) and 405 nm light-emitting diode (LED) ($Ca^{2+}$-independent, purple) was captured. 465 and 405 nm traces were scaled and subtracted to remove motion artifacts and photobleaching (blue). Fluorescence was then normalized to the whole recording session and represented as a *z*-score (black). (**C**) *Top*: Heat maps show the average NAc dopamine release on each trial (row) throughout the 30-trial session (trial 1 at the top). On each trial, 200 µl of 0.3 M sucrose (top panel) or 0.001 M quinine (bottom panel) was delivered over 5 s. *Bottom*: Average dopamine release averaged across all trials aligned to the onset of intraoral delivery. Dotted line represents onset of infusion and gray shading reflects infusion duration and time window for statistical analysis. *Inset*: z-score averaged first across the infusion period and then across trials and rats. Individual points represent data from each rat and lines connect sucrose (green) and quinine (orange) data for each rat. (**D**) Relative frequency histogram of dopamine responses (mean *z*-score) to sucrose and quinine from every trial reported. (**E**) Receiver operating characteristic (ROC) of (**D**) determined a discriminable difference between dopamine responses. Scale bars in (**B**): 10 s (465 and 405), 5 ΔF/F/10 s (subtraction), 1 *z*-score/10 s (normalized). Mean ± SEM are represented as solid lines and shading (**C**); *p < 0.05, paired *t*-test.

The online version of this article includes the following source data and figure supplement(s) for figure 1:

**Source data 1.** Quantified dopamine responses to intraoral sucrose and quinine.

**Figure supplement 1.** NAc lateral shell recording placements from primary taste experiment.

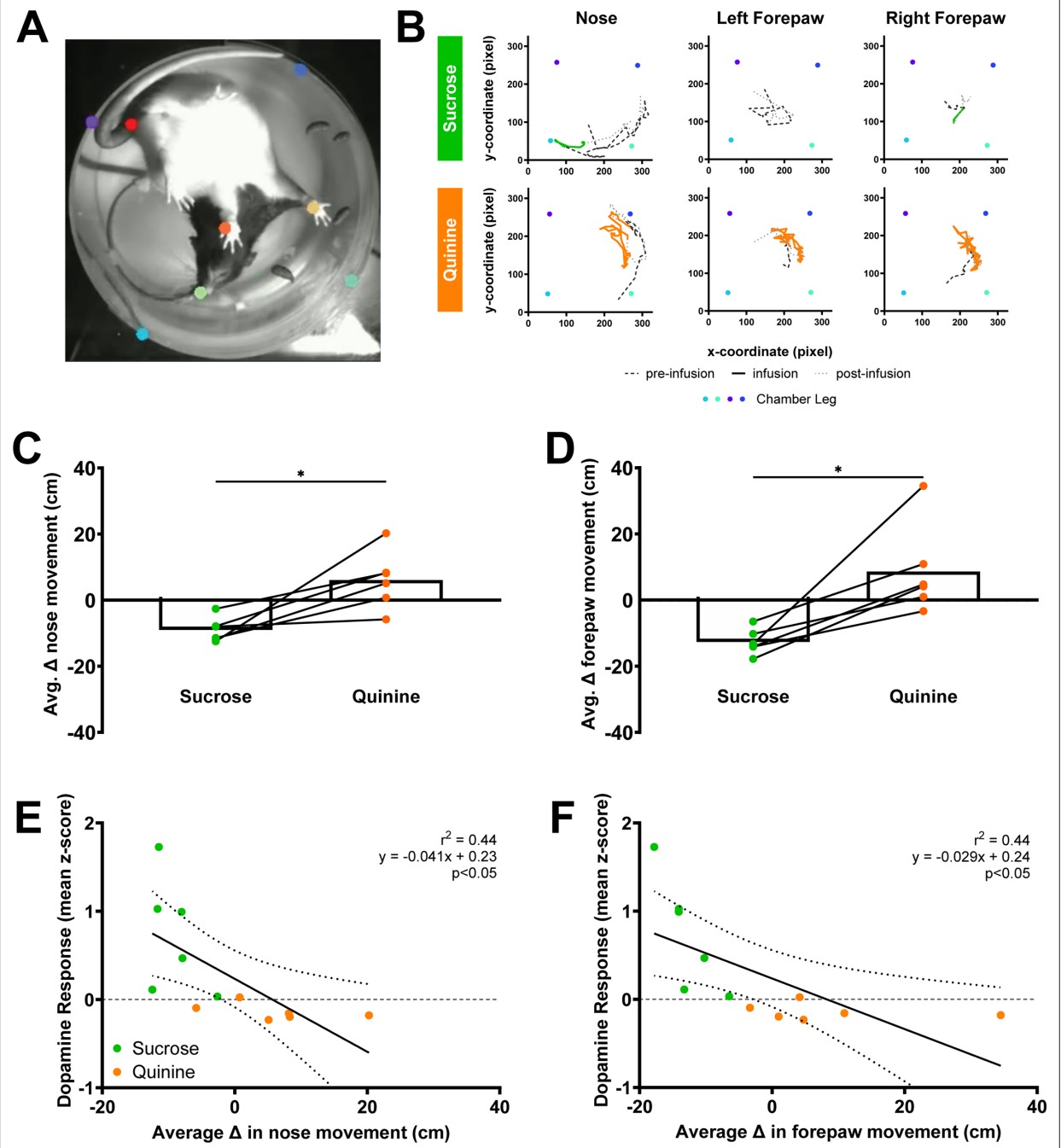

**Figure 2.** Aversive taste stimuli are linked to increased movement of nose and forepaws. (**A**) Representative image of nose, forepaws, tail base, and chamber legs tracking of a rat in a cylindrical chamber from a below chamber perspective. Positional coordinates of selected features were obtained using a model created via DeepLabCut, an open-source deep-learning pose estimation program. Custom MATLAB scripts were used to analyze movement from the positional data. (**B**) Representative movement of nose and forepaws tracked by a DeepLabCut model during 5 s pre-infusion, sucrose or quinine infusion, and post-infusion periods. (**C, D**) Behavioral reactivity was measured as the average change in nose movement or forepaw movement from baseline to infusion period. Intraoral infusion of quinine produces a greater behavioral reactivity. (**E, F**) Relationship between average change in behavioral reactivity and mean z-score of NAc dopamine during infusion averaged by session. Data in (**C, D**) are represented as means; *$p < 0.05$, paired t-test. Lines in (**E, F**) denote the linear relationship between parameters with dotted lines as 95% confidence intervals. p-value of linear regressions indicates slope's deviation from zero.

The online version of this article includes the following source data for figure 2:

**Source data 1.** Normalized nose and forepaw movements to intraoral sucrose and quinine and associated dopamine responses.

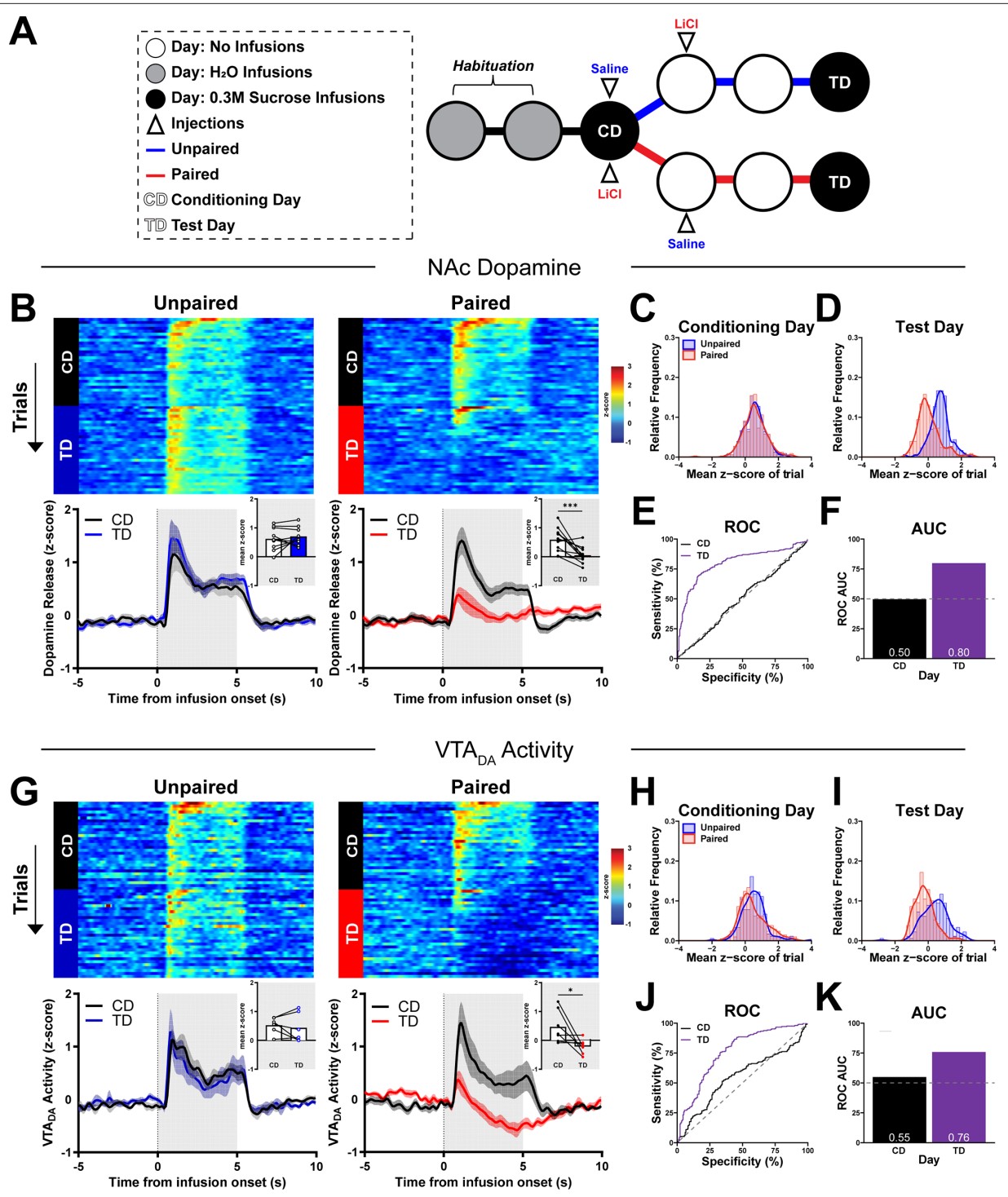

**Figure 3.** Pairing of lithium chloride (LiCl)-induced malaise to sucrose suppresses phasic dopamine responses to intraoral sucrose delivery. (**A**) Schematic of Single-pairing conditioned taste aversion (CTA) timeline. Subjects were first habituated to 30 brief intraoral infusions (200 µl/trial) of water at varying intertrial intervals (35–55 s) daily for 2 days. On Conditioning Day (CD), rats received intraoral infusions of 0.3 M sucrose and were then injected i.p. with saline (Unpaired) or 0.15 M LiCl (Paired). Rats received the counterbalanced injection in the home cage the next day and were untreated the following day. On Test Day (TD), rats received intraoral infusions (parameters identical to CD). (**B**) NAc dopamine across trials and sessions before and after intraoral sucrose delivery onset. *Top*: Heat maps show average NAc dopamine release on each trial (row) throughout the 30-trial session (trial 1 at the top) during both CD and TD. *Bottom*: Average dopamine release averaged across all trials aligned to the onset of intraoral delivery of sucrose on CD and TD. *Inset*: z-score averaged first across the infusion period and then across trials and rats. Individual points represent data from each rat, and lines connect CD and TD data for each rat. (**C, D**) Relative frequency histogram of dopamine release responses to sucrose on CD and TD for

*Figure 3 continued on next page*

*Figure 3 continued*

every trial reported as mean *z*-score for both Unpaired and Paired subjects. (**E**) Receiver operating characteristic (ROC) of relative frequency distributions of mean *z*-score acquired from each trial of on CD and TD between treatment groups. (**F**) Plotted area under the curve (AUC$_{ROC}$) values of E. (**G–K**) Recordings of VTA$_{DA}$ activity from Unpaired and Paired rats reported with same conventions as **B–F**. Data in **B** and **G** are represented as mean ± SEM; *p < 0.05, ***p < 0.005, paired *t*-test.

The online version of this article includes the following source data and figure supplement(s) for figure 3:

**Source data 1.** Quantified NAc dopamine and VTA$_{DA}$ activity responses to intraoral sucrose before and after CTA formation.

**Figure supplement 1.** NAc lateral shell and VTA recording placements from Single-pairing conditioned taste aversion (CTA) experiment.

the Unpaired rats, nose movement reactions were comparable between days (p > 0.05; *Figure 4A*) but Paired subjects showed increased nose movement to intraoral infusions from TD relative to CD (p = 0.0009; *Figure 4A*). Analysis of forepaw movement also indicated differences between Treatment groups (main effect of Treatment: $F_{(1,18)}$ = 8.81, p < 0.01; *Figure 4B*). Forepaw movement did not change from CD to TD in Unpaired rats (p > 0.05; *Figure 4B*) but increased in Paired rats (p < 0.05; *Figure 4B*). Further, Paired rats showed greater forepaw reactivity than Unpaired rats on TD (p < 0.01; *Figure 4B*). Examining the relationship between dopamine release and behavioral reactivity revealed negative correlations between dopamine release and nose movement ($r^2$ = 0.26, slope = −0.025, p < 0.001; *Figure 4C*) and between dopamine release and forepaw moment ($r^2$ = 0.22, slope = −0.021, p < 0.005; *Figure 4D*).

Similar results were observed in the separate group of rats from which VTA$_{DA}$ activity recordings were made. Intraoral infusions of sucrose elicited different nose movement responses between subjects across testing days (Treatment × Day: $F_{(1,10)}$ = 6.60, p < 0.05; *Figure 4E*). Specifically, nose movement was comparable between CD and TD in Unpaired rats (p > 0.05; *Figure 4E*) but significantly greater on TD relative to CD in Paired rats (p < 0.005; *Figure 4E*). Forepaw movement responses between subjects and across testing days were similar (Treatment × Day: $F_{(1,10)}$ = 6.34, p < 0.05; *Figure 4F*). Forepaw movement did not change across testing days in Unpaired rats (p > 0.05; *Figure 4F*) but increased in Paired rats (p < 0.005; *Figure 4F*). Further, dopamine activity was also negatively correlated to nose movement ($r^2$ = 0.48, slope = −0.057, p < 0.0005; *Figure 4G*) and forepaw moment ($r^2$ = 0.51, slope = −0.044, p < 0.0001; *Figure 4H*).

## Repeated 'safe' re-exposure to sucrose reduces the suppression of sucrose-evoked phasic VTA$_{DA}$ activity in parallel with behavioral extinction

A CTA can be extinguished when the conditioned stimulus (CS, i.e., intraoral sucrose) is repeatedly presented in the absence of the unconditioned stimulus (US, i.e., malaise induced by LiCl injection) (*Hadamitzky et al., 2015*); however, it is unclear if CTA-suppressed dopamine responses to intraoral sucrose recover with extinction. Here, VTA$_{DA}$ activity was recorded (*Figure 5—figure supplement 1*) in a separate cohort of subjects that underwent Single-pairing CTA followed by five consecutive Extinction sessions, where intraoral sucrose was presented without subsequent injection (*Figure 5A*). To examine voluntary sucrose preference or avoidance, rats were given access to sucrose and water for 2 hr following each intraoral sucrose session (*Figure 5A*). In a within-subjects comparison, intraoral sucrose infusions produced comparable VTA$_{DA}$ activity responses in Unpaired rats across CD and all Extinction days (E1–E5, Unpaired: main effect of session: $F_{(2.60,15.59)}$ = 2.06, p > 0.05; *Figure 5B, C*). In contrast, VTA$_{DA}$ responses to intraoral sucrose were influenced by CTA formation and extinction (Paired: main effect of Day: $F_{(2.18,15.27)}$ = 10.32, p < 0.005; *Figure 5B, C*). CTA formation suppressed sucrose-driven VTA$_{DA}$ responses (Paired: CD vs. E1, p < 0.005; *Figure 5B, C*), but VTA$_{DA}$ responses recovered to CD levels with re-exposure (Paired: CD vs. E2, p > 0.05; *Figure 5B, C*). To assess between-subject differences across Days, we used an ROC to compare the distribution of VTA$_{DA}$ responses on individual trials between Treatments (Unpaired vs. Paired). VTA$_{DA}$ responses to intraoral sucrose were discriminable on E1 (AUC$_{ROC}$ = 0.76) and E2 (AUC$_{ROC}$ = 0.77; *Figure 5E, F*) but not during E3–E5 (AUC$_{ROC}$ ≤0.63).

Preference scores also reflected changing responses to sucrose. Paired rats showed an avoidance of sucrose relative to Unpaired rats (main effect of group $F_{(1,13)}$ = 35.24, p < 0.0001; *Figure 6A*) and sucrose preference scores of Paired rats changed across Extinction days (Treatment × Day: $F_{(4,52)}$ =

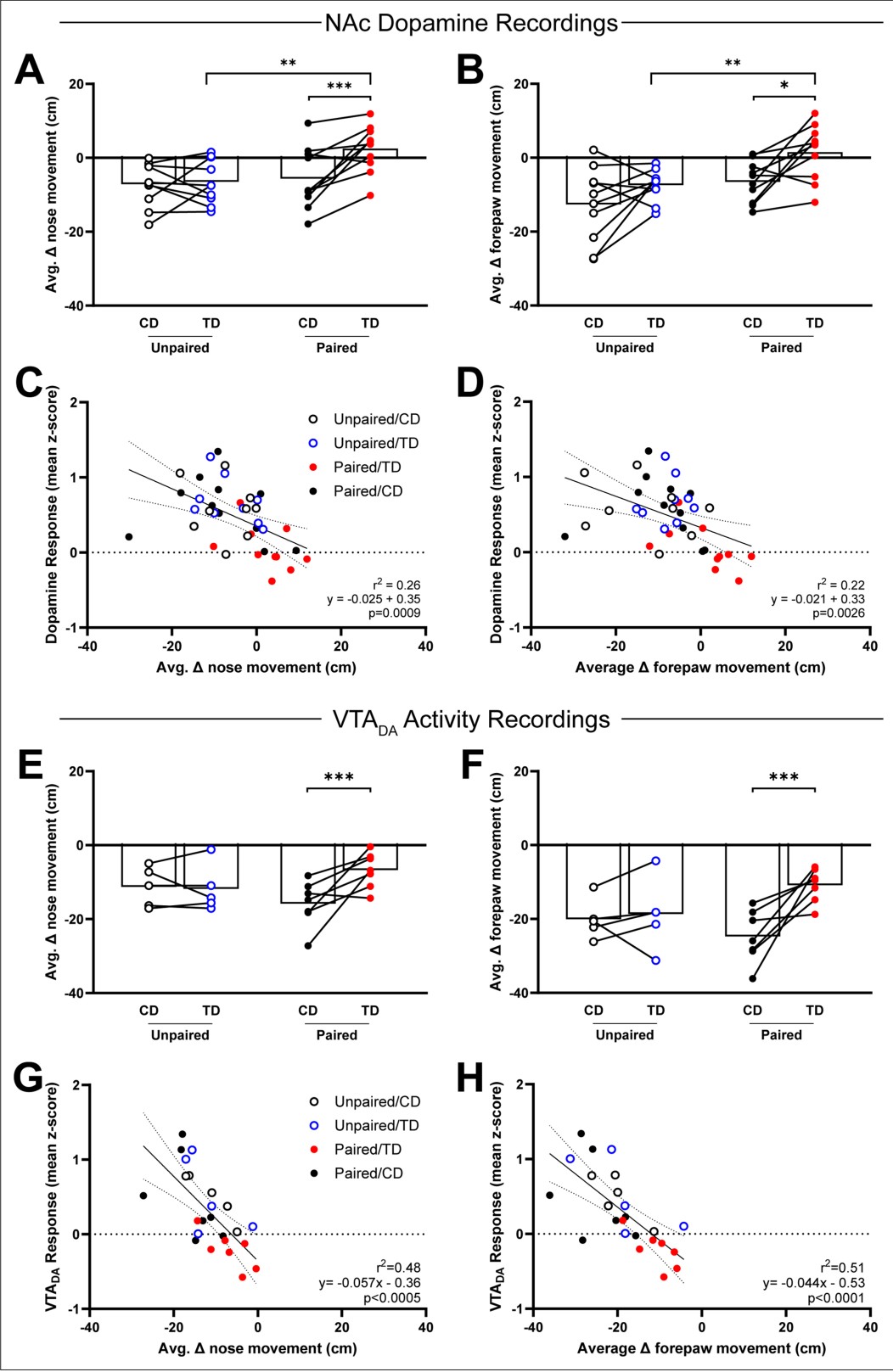

**Figure 4.** Suppressed dopamine responses correlate with enhanced behavioral reactivity to intraoral sucrose delivery after conditioned taste aversion (CTA) formation. (**A, B**) Behavioral reactivity was quantified as the change in movement from baseline to infusion period. In Unpaired subjects, the average behavioral reactivity of nose (*left*) or forepaw (*right*) movement did not change from Conditioning Day (CD) to Test Day (TD). In Paired subjects,

*Figure 4 continued on next page*

*Figure 4 continued*

behavioral reactivity increased from CD to TD. (**C**) Relationship between behavioral reactivity of nose movement and mean *z*-score of NAc dopamine during infusion averaged by session. (**D**) Relationship between average change in behavioral reactivity of forepaw movement and mean *z*-score of $VTA_{DA}$ activity responses during infusion averaged by session. Data in **A**, **B**, **E**, and **G** are represented as means; *p < 0.05, **p < 0.01; ***p < 0.005, two-way RM ANOVA with Uncorrected Fisher's LSD post hoc. Line in **C**, **D**, **F**, and **H** denotes the linear relationship between parameters with dotted lines as 95% confidence intervals. p-value of linear regressions indicate slope's deviation from zero.

The online version of this article includes the following source data for figure 4:

**Source data 1.** Normalized nose and paw movements to intraoral sucrose before and after CTA formation and associate dopamine responses.

33.14, p < 0.0001; *Figure 6A*). Relative to Unpaired rats, Paired subjects had a significantly lower preference for sucrose on E1 (p < 0.0001; *Figure 6A*) and E2 (p < 0.0001; *Figure 6A*), which recovered to Unpaired levels by E3 (p > 0.05; *Figure 6A*). Importantly, sucrose preference was predicted by the earlier recording of $VTA_{DA}$ responses to intraoral sucrose ($r^2 = 0.38$, slope = 0.33, p < 0.0001; *Figure 6B*).

To confirm the recovery of $VTA_{DA}$ responses after CTA extinction was due to presentations of intraoral sucrose without subsequent US exposure (malaise) and not simply the passage of time, we recorded $VTA_{DA}$ activity from a separate group of rats using the Single-pairing CTA paradigm but with a delayed TD (*Figure 7—figure supplement 1*). Here, rats were re-exposed to intraoral sucrose 7 days after CD (*Figure 7A*); thus, the timing of TD was equated with E5 from the Single-pairing paradigm with Extinction (*Figure 5A*). Intraoral sucrose evoked comparable $VTA_{DA}$ responses in Unpaired rats between CD and TD (Unpaired: $t(4) = 0.97$, p > 0.05; *Figure 7B*). In contrast, dopamine responses were significantly suppressed in Paired rats on the delayed TD relative to CD (Paired: $t(4) = 10.24$, p = 0.0005; *Figure 7B*). Further, while individual $VTA_{DA}$ responses to intraoral sucrose in Unpaired and Paired rats could not be discriminated on CD ($AUC_{ROC} = 0.59$), they were well discriminated on TD ($AUC_{ROC} = 0.84$, *Figure 7C–F*).

## Suppression of phasic dopamine responses in CTA scales with conditioning and extinction

To determine if phasic dopamine responses scale with the strength of a taste aversion, $VTA_{DA}$ activity was monitored in a separate cohort of rats undergoing a Repeated-pairing CTA with Extinction paradigm (*Figure 8—figure supplement 1*). Here, three CD cycles with accompanying non-contingent injection and 'off" days (C1–C3) were administered to Unpaired and Paired rats followed immediately by Extinction days E1–E8. In addition, sucrose preference tests were conducted after each Extinction day (*Figure 7A*). In Unpaired rats, average $VTA_{DA}$ activity was comparable across all Conditioning (C1–C3) and Extinction days (E1–E8, Unpaired: main effect of day: $F_{(3.88,23.30)} = 1.99$, p > 0.05; *Figure 8B–D*). In contrast, Paired rats showed significant changes in $VTA_{DA}$ responses to sucrose across the paradigm (Paired: main effect of day: $F_{(2.86,19.98)} = 16.07$, p < 0.0001; *Figure 8B–D*). In Paired rats, $VTA_{DA}$ responses were suppressed on C3 through E4 (Paired: Relative to C1: C3–E4, p < 0.05; *Figure 8B–D*) but not subsequent Extinction days (Paired: Relative to C1: E5–E8, p > 0.05; *Figure 8B–D*). Between-subject differences were also evident; responses to intraoral sucrose were discriminable between Unpaired and Paired groups on C3 ($AUC_{ROC} = 0.88$) and E1–E6 ($AUC_{ROC} > 0.7$; see *Figure 8E–G* for individual $AUC_{ROCs}$).

Behavioral reactivity was simultaneously recorded with $VTA_{DA}$ activity. Intraoral sucrose delivery produced comparable behavioral responses in Unpaired rats across all days (Unpaired: main effect of day: $F_{(3.19,19.12)} = 0.74$, p > 0.05; *Figure 9A*). In contrast, Paired subjects showed shifts in behavioral responses across conditioning and extinction days (Paired: main effect of day: $F_{(4.02,28.15)} = 7.70$, p = 0.0003; *Figure 9A*). Relative to the first conditioning day, head movement was elevated from C3 through E3 and decreased with subsequent extinction sessions (Paired: Relative to C1: C3, E2, E3 p < 0.05, E1 p < 0.005; *Figure 9A*). $VTA_{DA}$ activity and behavioral reactivity were negatively correlated across all rats and days ($r^2 = 0.43$, slope = −9.52, p < 0.0001; *Figure 9B*).

After each extinction session, we conducted a sucrose preference test. Paired rats avoided sucrose compared to Unpaired rats (main effect of group: $F_{(1,13)} = 59.78$, p < 0.0001; *Figure 9C*). Specifically,

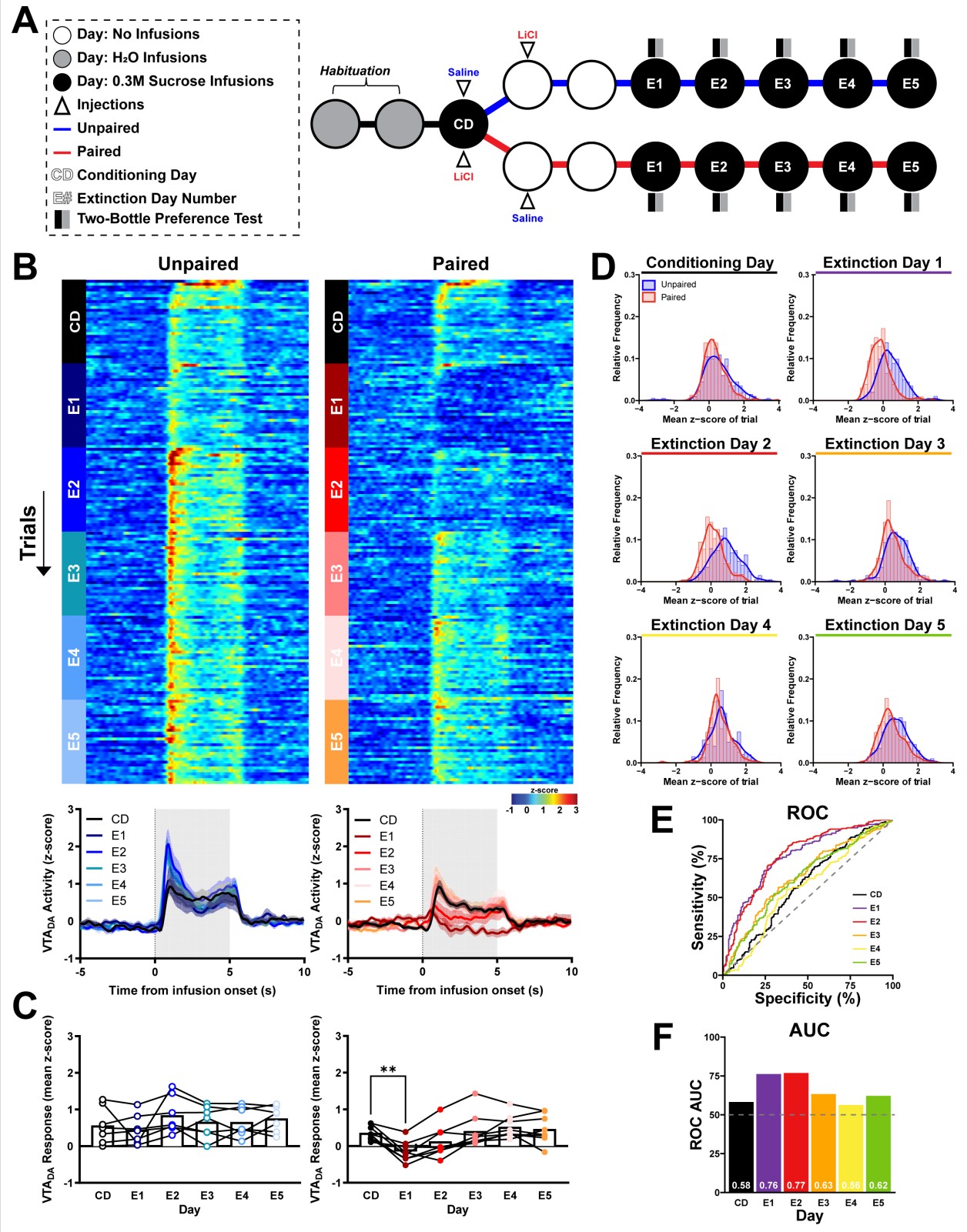

**Figure 5.** Sucrose exposure under extinction conditions ameliorates conditioned taste aversion (CTA)'s suppression of phasic dopamine responses to intraoral sucrose. (**A**) Schematic of the CTA paradigm consisting of a ingle-pairing of lithium chloride (LiCl) or saline pairing to sucrose followed by five consecutive sessions intraoral sucrose not subject to additional US (malaise) exposure. CTA training as in *Figure 3* was conducted followed by five Extinction sessions (E1–E5), in which rats received intraoral infusions (parameters identical to Test Day [TD]) without any additional injections. (**B**) VTA$_{DA}$

*Figure 5 continued on next page*

*Figure 5 continued*

across trials and sessions before and after intraoral sucrose delivery onset. *Top*: Heat maps show average VTA$_{DA}$ activity on each trial (row) throughout the 30 trials (trial 1 at the top) during the Conditioning Day (CD) and E1–E5 sessions. *Bottom*: Average VTA$_{DA}$ activity averaged across all trials and aligned to the onset of intraoral delivery of sucrose. (**C**) *z*-score averaged first across the infusion period and then across trials and rats. Individual points represent data from each rat and lines connect CD and E1–E5 data for each rat. (**D**) Relative frequency histogram of VTA$_{DA}$ responses to sucrose on all test days for every trial reported as mean *z*-score for both Unpaired and Paired subjects. (**E**) Receiver operating characteristic (ROC) of relative frequency distributions of mean *z*-score acquired from each trial of on all test sessions between treatment groups. (**F**) Plotted area under the curve (AUC$_{ROC}$) values of (**E**). Data in (**B**) are represented as mean ± SEM; **p < 0.01, one-way RM ANOVA with Dunnett's multiple comparisons test post hoc.

The online version of this article includes the following source data and figure supplement(s) for figure 5:

**Source data 1.** Quantified VTA$_{DA}$ activity responses to intraoral sucrose across CTA extinction.

**Figure supplement 1.** VTA recording placements from Delayed-test conditioned taste aversion (CTA) experiment.

Paired rats had a lower sucrose preference on E1 through E5 (E1 and E2 p < 0.0001, E3 p < 0.005, E4 p = 0.0001, E5 p < 0.05; *Figure 9C*). Similar to behavioral reactivity but with more predictive power, VTA$_{DA}$ responses were tightly correlated with post-session sucrose preference (*r*$^2$ = 0.59, slope = 0.48, p < 0.0001; *Figure 9D*).

# Discussion

Unexpected rewards and reward-predictive cues evoke phasic dopamine activity and NAc dopamine release (*Schultz, 1998*). Responses to aversion remain more controversial (*McCutcheon et al., 2012*; *Morales and Margolis, 2017*) and may differ based on sensory transduction pathways (*McCutcheon et al., 2012*) and whether the aversive outcome can be avoided (*Goedhoop et al., 2022*; *Oleson et al., 2012*) – which contribute to competing hypotheses that phasic dopamine responses signal either valence (reward vs. aversion) or salience (reward and aversion *Kutlu et al., 2021*). Here, we circumvent common obstacles to determining responses to aversive stimuli by capturing real-time dopamine responses to the same taste stimulus – sucrose – and modulating its valence through conditioning. Intraoral delivery of sucrose eliminated avoidance as a potential confound. Measurement of behavior ensured that rats were responding to the intraoral infusion, eliminating differences in stimulus salience as a contributing factor. A key aspect of our paradigms is that all rats had identical exposure to sucrose and illness with the temporal proximity of sucrose and illness being the lone difference. Ultimately, we show that dopamine responses to sucrose scale with valence across acquisition and extinction of a CTA.

## Phasic dopamine activity differentially encodes the valence of primary appetitive and aversive taste stimuli

Dopamine responses to primary aversive stimuli are mixed. Noxious stimuli have been reported to increase the firing rate of dopamine cell bodies in the VTA (*Anstrom et al., 2009*; *Anstrom and Woodward, 2005*) and dopamine release in the NAc (*Budygin et al., 2012*; *Kiyatkin, 1995*; *Mikhailova et al., 2019*; *Young, 2004*; *Young et al., 1993*). Conversely, others have reported that noxious stimuli inhibit VTA$_{DA}$ firing (*Brischoux et al., 2009*; *Ungless et al., 2004*) and NAc dopamine release (*Mantz et al., 1989*; *Wenzel et al., 2015*) or that responses vary depending on the VTA$_{DA}$ subpopulation or dopamine release site (*Brischoux et al., 2009*; *de Jong et al., 2019*; *Zweifel et al., 2011*). Aversive stimuli can drive excitatory inputs to the VTA (*Amo et al., 2024*; *Faget et al., 2024*). With respect to taste stimuli specifically, some reports indicate decreased (*Roitman et al., 2008*; *Twining et al., 2015*), while others report increased (*Bassareo et al., 2002*; *Kutlu et al., 2021*) dopamine responses to bitter quinine. Here, we used intraoral delivery of appetitive and aversive taste stimuli to circumvent confounds including active or passive avoidance, and food or water deprivation. Recordings were targeted to the lateral VTA and the corresponding approximate terminal site in the NAc lateral shell (*Lammel et al., 2008*). Subregional differences in dopamine activity likely contribute to mixed findings on dopamine and affect. For example, dopamine in the NAc lateral shell differentially encodes cues predictive of rewarding sucrose and aversive footshock, which is distinct from NAc medial shell dopamine responses (*de Jong et al., 2019*). Our findings are similar to prior work from our group targeting recordings of the NAc dorsomedial shell (*Hsu et al., 2020*; *McCutcheon et al., 2012*; *Roitman et al., 2008*): there, intraoral sucrose increased NAc dopamine release while the

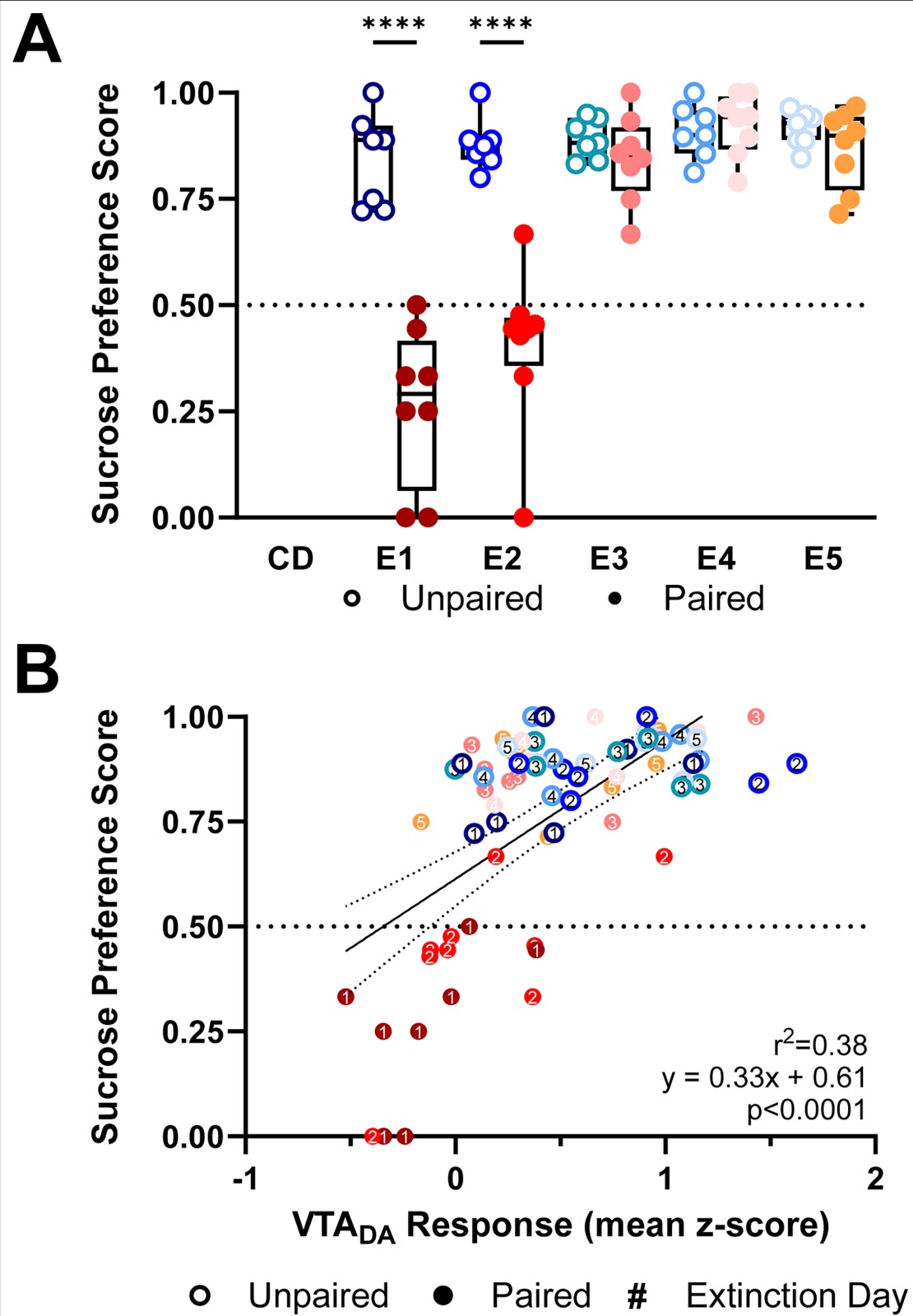

**Figure 6.** Suppressed dopamine responses to sucrose predict conditioned taste avoidance. (**A**) Each Extinction session was followed by a 2-hour Two-Bottle Preference test with access to both sucrose and water. Sucrose preference scores were calculated as the percent of sucrose solution consumed from the sum of sucrose and water consumed. Paired (red, closed circles) rats showed decreased sucrose preference on E1 and E2 relative to Unpaired rats (blue, open circles). (**B**) Average VTA$_{DA}$ responses during intraoral sucrose delivery were positively correlated to sucrose preference scores calculated

*Figure 6 continued on next page*

*Figure 6 continued*

from the Two-Bottle Preference test. Data in (**A**) are represented as means; ****p < 0.001, two-way RM ANOVA with Šidák multiple comparisons test post hoc. Line in (**B**) denotes the linear relationship between parameters with dotted lines as 95% confidence intervals. p-value of linear regression indicates slope's deviation from zero.

The online version of this article includes the following source data for figure 6:

**Source data 1.** Sucrose preference scores and average VTA$_{DA}$ activity responses to sucrose across CTA extinction.

response in the same rats to quinine was significantly lower. Neuroanatomical data support projections to the VTA from relatively early nodes of the gustatory pathway (e.g., parabrachial nucleus; PBN) and specifically sucrose and quinine-responsive neurons in the PBN (*Boughter et al., 2019*). It remains to be determined if sucrose- and quinine-responsive PBN neurons synapse onto different VTA cell types (e.g., dopamine neurons for the former and GABA neurons for the latter). Regardless, it is

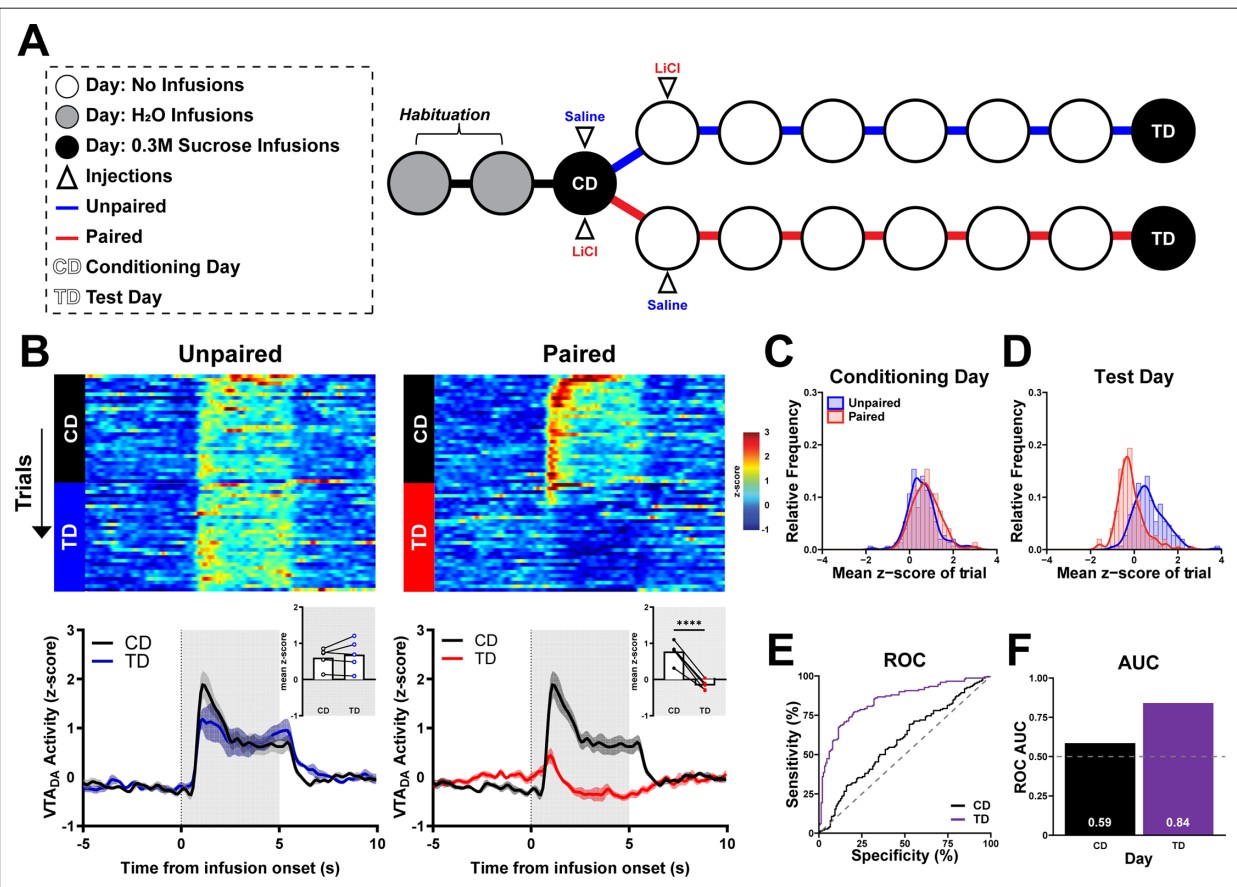

**Figure 7.** Delayed testing after sucrose pairing to lithium chloride (LiCl)-induced malaise suppresses dopamine response to sucrose. (**A**) Schematic of the Delayed-test conditioned taste aversion (CTA) paradigm. CTA training as in *Figure 3* was conducted. To match the timeline of Single-pairing CTA with Extinction, rats had five additional untreated days followed by Test Day (TD), which corresponded to the delay from Conditioning Day (CD) to E5. On TD, rats received intraoral infusions (parameters identical to E1). (**B**) VTA$_{DA}$ across trials and sessions before and after intraoral sucrose delivery onset. *Top*: Heat maps show average VTA$_{DA}$ activity on each trial (row) throughout the 30-trial session (trial 1 at the top) during CD and TD sessions. *Bottom*: Average VTA$_{DA}$ activity averaged across all trials aligned to the onset of intraoral delivery of sucrose. *Inset*: z-score averaged first across the infusion period and then across trials and rats. Individual points represent data from each rat and lines connect CD TD data for each rat. (**C, D**) Relative frequency histogram of dopamine release responses to sucrose on CD and TD for every trial reported as mean z-score for both Unpaired and Paired subjects. (**E**) Receiver operating characteristic (ROC) of relative frequency distributions of mean z-score acquired from each trial of on CD and TD between treatment groups. (**F**) Plotted area under the curve (AUC$_{ROC}$) values of (**E**). Data in (**B**) are represented as mean ± SEM; ****p < 0.001, paired t-test.

The online version of this article includes the following source data and figure supplement(s) for figure 7:

**Source data 1.** Quantified VTA$_{DA}$ activity to intraoral sucrose across the Delayed-test CTA paradigm.

**Figure supplement 1.** VTA recording placements from Single-pairing conditioned taste aversion (CTA) with Extinction experiment.

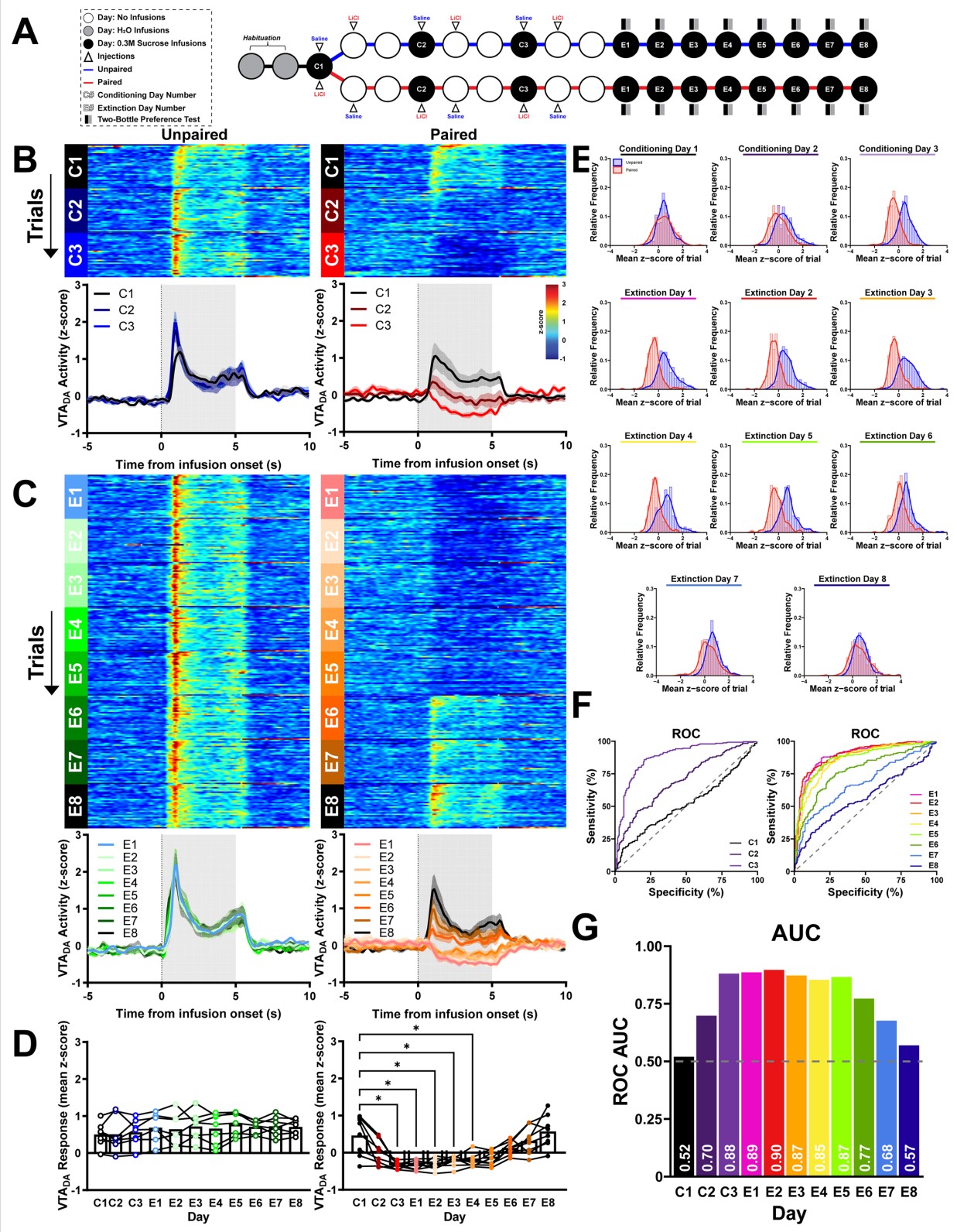

**Figure 8.** Phasic dopamine responses to intraoral sucrose scale to the strength of the conditioned taste aversion (CTA). (**A**) Schematic of the Repeated-pairing CTA paradigm. The 3-day CTA conditioning process (as in **Figure 3**) was repeated three times (C1–C3). Following conditioning, subjects received eight daily Extinction sessions (E1–E8) with intraoral infusions of sucrose followed by no US (malaise) exposure. After the behavioral sessions, rats had access to sucrose and water consumption for 2 hr (Two-Bottle Preference test). (**B**) VTA$_{DA}$ across trials and sessions before and after intraoral

*Figure 8 continued on next page*

*Figure 8 continued*

sucrose delivery onset. *Top*: Heat maps show average VTA$_{DA}$ activity on each trial (row) throughout the 30 trial sessions (trial 1 at the top) during the three conditioning sessions (C1–C3). *Bottom*: Associated average traces of VTA$_{DA}$ activity averaged across all trials and aligned to the onset of intraoral delivery of sucrose. (**C**) Conventions as in (**B**) but for all eight non-reinforced sucrose sessions (E1–E8). (**D**) z-score averaged first across the infusion period and then across trials and rats for all conditioning and subsequent test sessions. Individual points represent data from each rat and lines connect C1–C3 and E1–E8 data for each rat. (**E**) Relative frequency histogram of VTA$_{DA}$ responses to sucrose on all test days for every trial reported as mean z-score for both Unpaired and Paired subjects. (**F**) Receiver operating characteristic (ROC) of relative frequency distributions of mean z-score acquired from each trial of on all conditioning (left) and extinction (right) days between treatment groups. (**G**) Plotted area under the curve (AUC$_{ROC}$) values of (**F**). Data in (**B**) and (**D**) are represented as mean ± SEM;; *p < 0.05, one-way RM ANOVA with Dunnett's multiple comparisons test post hoc.

The online version of this article includes the following source data and figure supplement(s) for figure 8:

**Source data 1.** Quantified VTA$_{DA}$ activity to intraoral sucrose across repeated pairings.

**Source data 2.** Quantified VTA$_{DA}$ activity to intraoral sucrose across repeated extinction sessions.

**Source data 3.** Quantified averaged VTA$_{DA}$ activity to intraoral sucrose across repeated parings and extinction sessions.

**Source data 4.** Data for ROC analyses across conditioning and extinction sessions.

**Figure supplement 1.** VTA recording placements from Repeated-pairing conditioned taste aversion (CTA) with Extinction experiment.

important to recognize that affective coding of stimuli may differ across VTA and NAc subregions and in response to different classes of stimuli (e.g., pain, taste).

Differences in stimulus salience could contribute to the dopamine responses observed here. Indeed, recent work has supported dopamine alignment with the encoding of stimulus salience (*Chen and Bruchas, 2021*; *Kutlu et al., 2021*) to contribute to prioritizing attentional processes toward stimuli of high intensity. Further, human VTA activation is greater for novel, more highly salient stimuli than to familiar stimuli (*Bunzeck and Düzel, 2006*). Here, behavioral responses to quinine were greater than those for sucrose supporting quinine as a highly salient stimulus. Nonetheless, the strong inverse relationship between behavioral reactivity and dopamine release observed here, and particularly the lack of a dopamine response to quinine refutes the idea that phasic dopamine encodes the salience of the stimulus.

## Phasic dopamine activity differentially encodes appetitive versus aversive sucrose

The differential dopamine response to appetitive and aversive tastes supports affective encoding by the mesolimbic system. However, sucrose and quinine act on different taste receptors (*Schier and Spector, 2019*) and may access the VTA via parallel pathways (*Boughter et al., 2019*). We therefore employed a CTA paradigm to measure dopamine responses to the same stimulus when the stimulus was appetitive versus aversive. While dopamine responses were identical across groups on CD (when rats were initially naive to intraoral sucrose), pairing intraoral sucrose infusions with LiCl injection caused changed, suppressed dopamine responses to intraoral sucrose when it was next administered (TD). Importantly, the temporal relationship between illness and initial sucrose exposure, as in seminal studies (*Smith and Roll, 1967*), determined both the formation of a CTA (as evidence in behavioral reactivity measures) and suppressed dopamine activity and release. Indeed, LiCl injection administered 24 hr after intraoral sucrose (Unpaired) failed to modulate dopamine signaling. Recent work has suggested that dopamine release from terminals within the NAc could result from intra-NAc processes that are independent of cell body activity (*Cachope and Cheer, 2014*; *Mohebi et al., 2023*; *Threlfell et al., 2012*). However, we found similar modulation of dopamine responses in Paired rats when recordings captured NAc dopamine release or VTA$_{DA}$ cell body activity. Our data are more consistent with the good agreement between dopamine cell body activity, release, and post-synaptic signaling reported by others (*Lee et al., 2020*). Local GABA neurons innervate and suppress VTA$_{DA}$ neurons through neurotransmission at GABA$_A$ receptors, which can pause VTA$_{DA}$ activity and downstream dopamine release in the NAc (*Mathon et al., 2005*; *Tan et al., 2012*; *van Zessen et al., 2012*). Understanding local GABA dynamics will be an intriguing direction to identify mechanisms for suppression of dopamine signaling to innately aversive tastes as well as those that have acquired aversion through conditioning.

Throughout our studies, we found an inverse relationship between dopamine release/activity and behavioral reactivity to intraoral infusions. Decreases in dopamine activity can lead to an uncoupling

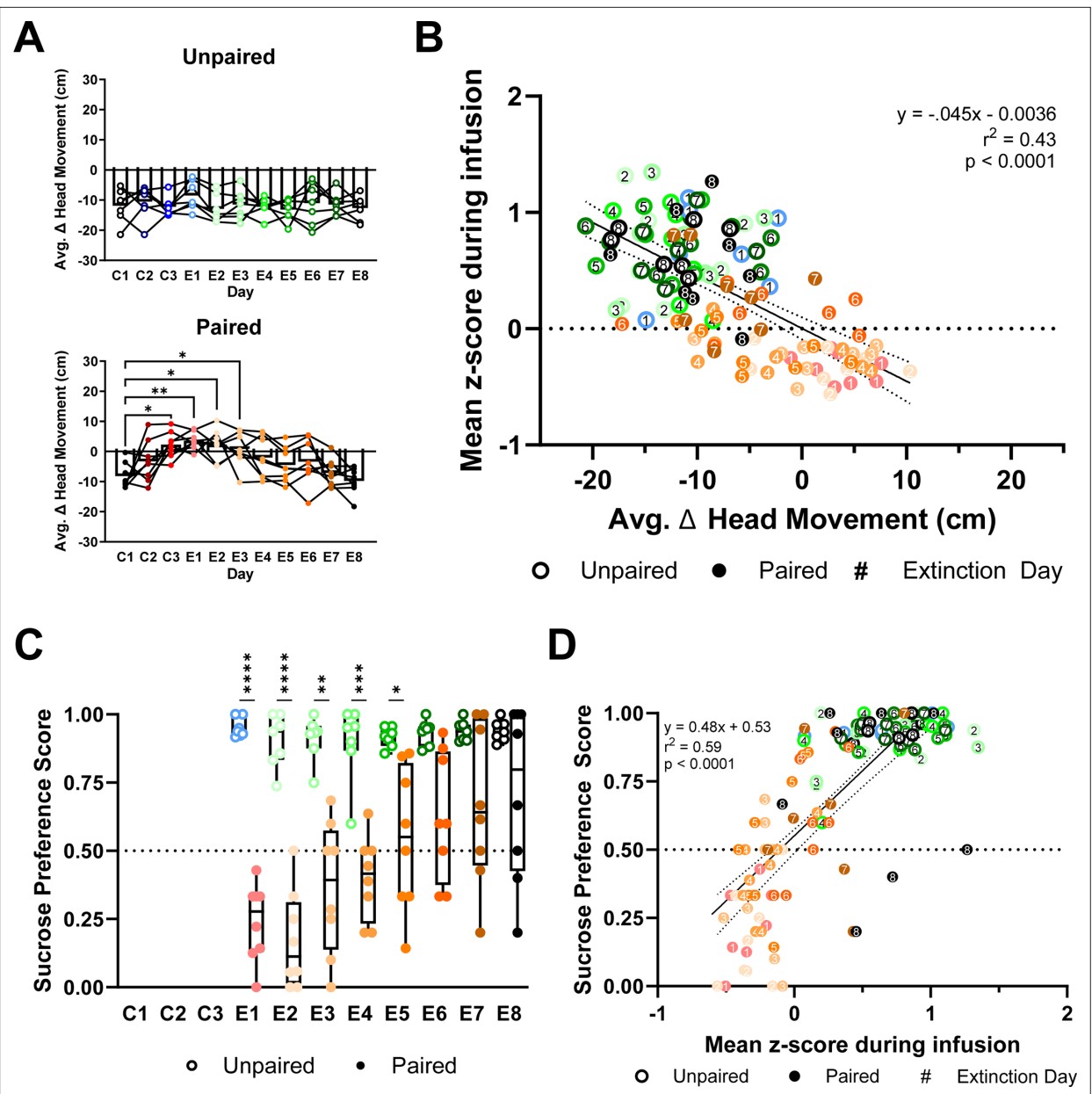

**Figure 9.** Intraoral sucrose-driven dopamine responses are negatively correlated to behavioral reactivity and predict sucrose preference. (**A**) Head movement to intraoral sucrose averaged across trials. *Top*: Unpaired rats showed comparable average change in head movement behavioral reactivity across testing sessions. *Bottom*: Paired subjects showed an increase in average head movement responses to intraoral sucrose from C1 to C2–E3. (**B**) Relationship between average head movement behavioral reactivity and $VTA_{DA}$ responses (in mean *z*-score) during infusion averaged by session. (**C**) After each E1–E8 session, rats were administered the Two-Bottle Preference test. Paired (red, closed circles) rats showed decreased sucrose preference on E1–E5 relative to Unpaired rats (blue, open circles). (**D**) Average $VTA_{DA}$ responses during intraoral sucrose delivery were positively correlated to sucrose preference scores calculated from the Two-Bottle Preference test. Data in (**A**) are represented as means; *$p < 0.05$, **$p < 0.01$, one-way RM ANOVA with Dunnett's multiple comparisons test post hoc. Data in (**C**) are represented as means; *$p < 0.05$, **$p < 0.01$, ***$p < 0.005$, ****$p < 0.001$, two-way RM ANOVA with Šidák multiple comparisons test post hoc. Line in (**B, D**) denotes the linear relationship between parameters with dotted lines as 95% confidence intervals. p-value of linear regression indicates slope's deviation from zero.

The online version of this article includes the following source data for figure 9:

**Source data 1.** Normalized head movement to intraoral infusions across repeated pairings and extinction sessions and associated $VTA_{DA}$ responses and sucrose preference scores.

of dopamine and high-affinity dopamine D2-like receptors expressed on medium spiny neurons (*Dreyer et al., 2010*). In turn, this could contribute to the different patterns of NAc activity reported for rewarding versus aversive primary (*Roitman et al., 2005*) and conditioned tastes (*Roitman et al., 2010*; *Wheeler et al., 2008*). As manipulations of NAc activity can influence taste reactivity (*Morales and Berridge, 2020*), differential dopamine responses to sucrose in rats with versus without CTA likely applies a critical filter for NAc processing in the service of oromotor output.

In our purely classical conditioning CTA paradigm, taste (i.e., intraoral sucrose) served as a conditioned stimulus for visceral malaise. Other aversive conditioning paradigms have shown development of decreased dopamine responses to cues predictive of primary aversive stimuli (*Goedhoop et al., 2022*; *Zhuo et al., 2024*). There are unique aspects of CTA worth emphasizing. As we've shown here, CTA can form with just one taste–illness pairing. In addition, CTA tolerates relatively long delays between the conditioned (taste) and unconditioned stimulus (LiCl). While LiCl was injected immediately after the conditioning session, this interval exceeded 30 min from the initial taste exposure. Moreover, indices of the malaise induced by LiCl do not emerge for many minutes, for example, ~5 min for lying on belly (*Aguilar-Rivera et al., 2020*), ~20 min for pica (*Aguilar-Rivera et al., 2020*), and ~20 min for decrease in core body temperature (*Bernstein et al., 1992*; *Guimaraes et al., 2015*). Thus, taste and illness memory traces must be integrated over many minutes. The PBN is essential for CTA formation (*Spector et al., 1992*) and contains neurons that process gustatory and illness information (*Carter et al., 2015*). The amygdala is also critical for CTA acquisition (*Gao et al., 2023*; *Inui et al., 2019*; *Morin et al., 2021*; *Morris et al., 1999*; *St. Andre and Reilly, 2007*). The same neurons within the basolateral nucleus of the amygdala are responsive to both novel taste and LiCl (*Barot et al., 2008*; *Zimmerman et al., 2024*). The rostral tegmental nucleus (RMTg) is a robust source of GABAergic input onto VTA$_{DA}$ neurons and suppresses their firing (*Balcita-Pedicino et al., 2011*; *Bourdy et al., 2014*). Primary aversive stimuli, including LiCl, increase RMTg neuronal firing (*Li et al., 2019*). Further, re-exposure to LiCl-paired saccharin increases cFos expression in the RMTg (*Glover et al., 2016*). Likewise, LiCl injections suppress electrically evoked NAc dopamine release (*Fortin et al., 2016*). How taste and illness traces and their integration ultimately modulate the mesolimbic system is an important ongoing direction.

## Suppression of dopamine signaling scales with CTA expression and extinction

Multiple conditioning trials caused greater suppression in dopamine activity evoked by the sucrose CS. This has been previously reported for audiovisual cues that predict discrete, noxious stimuli (e.g., air puff *Zhuo et al., 2024* and foot shock *Wilkinson et al., 1998*). Here, greater suppression occurred with each taste–illness pairing over just three conditioning trials spaced days apart. Taste, relative to other sensory stimuli, serves as a uniquely strong predictor for illness (*Garcia and Koelling, 1966*). After establishing CTA, repeated re-exposure to the conditioned stimulus without subsequent malaise drives CTA extinction (*Cantora et al., 2006*; *Mickley et al., 2004*; *Nolan et al., 1997*), allowing for new, contextual knowledge to conflict with the original association between the taste and negative post-ingestive outcome (*Bouton and Bolles, 1979*). We found that the suppressed dopamine response to sucrose gradually returned to conditioning day levels after repeated extinction sessions. The number of extinction sessions needed for the restoration of dopamine signaling scaled with the number of conditioning sessions (i.e., two extinction sessions for one pairing and five extinction sessions for three pairings). The return of dopamine responses to conditioning day levels was not simply due to the passage of time but rather required CS re-exposure. As with all experiments herein, dopamine responses were negatively correlated with behavioral reactivity across acquisition and extinction. Importantly, dopamine responses to intraoral sucrose were also predictive of home cage sucrose preference across extinction. The infralimbic (IL) cortex plays a role in extinction learning (*Quirk et al., 2006*). Stimulation of the IL suppresses aversive taste reactivity to sucrose after it had been paired with LiCl (*Hurley and Carelli, 2020*). Thus, while initial taste–LiCl associations may be due to subcortical (e.g., PBN, RMTg, amygdala) influences on sucrose-evoked VTA$_{DA}$ activity and NAc dopamine release, extinction of the suppressed dopamine response to sucrose may well engage top–down cortical processes.

Our studies support a role for dopamine in differentially encoding taste valence. It is important to recognize that there is considerable heterogeneity among dopamine neurons (*Phillips et al., 2022*).

Indeed, VTA neurons that release both glutamate and dopamine are important for aversion, and it is the dopamine signal from these neurons that is critical for this process (*Warlow et al., 2024*). The technical approaches used here to measure VTA$_{DA}$ activity and dopamine release in the NAc average responses across an unknown number of neurons. Thus, while our results suggest that most lateral VTA dopamine neurons decrease their activity and release of dopamine in the NAc lateral shell is lower to primary and conditioned aversive taste, subpopulations likely play different and critical roles in appetitive and aversive responses. Combining molecular markers and defining input/output relationships for VTA dopamine neurons will be critical for further advancing an understanding of dopamine and aversion. A range of clinically used drugs – from anti-obesity treatments to chemotherapeutics – induce malaise and can lead to the development of CTAs and a reduction in motivation. Thus, further understanding of how these associative processes influence dopamine signaling is important to yield therapeutic approaches with greater efficacy and improved adherence.

## Materials and methods

### Subjects

Male and female (intact, naturally cycling) Long Evans rats (>250 g) were bred from females heterozygous for expressing Cre recombinase under the control of the tyrosine hydroxylase promoter (TH:Cre) (Rat Research Resource Center, RRRC No. 659; *Witten et al., 2011*) and male wildtype rats (Charles River Laboratories). Offspring were genotyped for Cre (Transnetyx, Inc) via tissue from ear punches. TH:Cre+ ($n$ = 31 males, 22 females) rats were used for experiments where VTA$_{DA}$ activity was measured. TH:Cre− ($n$ = 13 males, 14 females) rats were used for experiments where NAc dopamine release was measured. Rats were individually housed in a temperature- and humidity-controlled room and on a 12:12 hr light:dark schedule (lights on 0700 hr). All experiments were conducted during the light cycle. Food and water were provided ad libitum. All studies were conducted in accordance with the National Institutes for Health Guide for the Care and Use of Laboratory Animals and approved by the Animal Care Committee (ACC) at the University of Illinois Chicago (ACC Protocol #: 22-119).

### Viruses

Adeno-associated viruses (AAVs) packaged with fluorescent sensors were used for in vivo fiber photometry. AAV1.hSyn.Flex.GCaMP6f.WPRE.SV40 (5 × 10$^{12}$ GC/ml, Addgene) was used to detect calcium (Ca$^{2+}$, *Chen et al., 2013*). AAV9.hSyn.GRAB_DA2h (1 × 10$^{13}$ GC/ml, Addgene) was used to detect dopamine (*Sun et al., 2020*).

### Surgeries

Rats were anesthetized with isoflurane, injected with an analgesic (1 mg/kg meloxicam, subcutaneous), and placed in a stereotaxic instrument. To record VTA$_{DA}$ activity, 1 µl of AAV1.hSyn.Flex.GCaMP6f.WPRE.SV40 was injected into the VTA (anterior/posterior [AP]: –5.40 mm, medial/lateral [ML]: +0.70 mm, dorsal/ventral [DV]: –8.15 mm, relative to Bregma) of TH:Cre+ rats at a rate of 0.1 µl/min. A 5-min post-injection period to allow for diffusion followed. A fiber optic (flat 400 µm core, 0.48 numerical aperture, Doric Lenses Inc) was then implanted just dorsal (DV: –8.00 mm) to the virus injection. To measure dopamine release, 1 µl of AAV9-hsyn-GRAB_DA2h was injected into the NAc lateral shell (AP: +1.5 mm, ML: +2.5 mm, DV: –8.0 mm relative to Bregma) of TH:Cre− rats. A fiber optic was implanted just dorsal to the virus injection (DV: –7.9 mm). For all surgeries, an intraoral catheter was also implanted. Catheters, made from ~6 cm length of PE6 tubing (Scientific Commodities, Inc) threaded through a Teflon washer, were inserted via hypodermic needle just lateral to the first maxillary molar. The needle was guided subcutaneously and exteriorized out of the incision at the top of the head. Implants were cemented to skull screws using Metabond (Parkell, Inc) and dental acrylic. To allow for recovery and construct expression, experiments began at minimum 21 days after surgery (*Figure 1A*).

### Fiber photometry recording

Light-emitting diodes (Doric Lenses) emitted 465 nm (Ca$^{2+}$ or dopamine-dependent) and 405 nm (Ca$^{2+}$-independent) wavelengths. At 30 µW emission power, frequencies of 465 and 405 nm light were sinusoidally modulated at 210 and 330 Hz, respectively. Light was coupled to a filter cube (FMC4,

Doric Lenses) and converged into an optical fiber patch cord mated to the fiber optic implant of the rat. Fluorescence was collected by the same fiber/patch cord and focused onto a photoreceiver (Visible Femtowatt Photoreceiver Model 2151, Newport). A lock-in amplifier and data acquisition system (RZ10X; Tucker Davis Technologies), was used to demodulate the fluorescence due to 465 and 405 nm excitation. Transistor–transistor logic (TTL) signals from the system operating behavioral events (Med-Associates, Inc) were sent to the data acquisition system, timestamped, and recorded along with fluorescence using software (Synapse Suite, Tucker Davis Technologies).

After acquisition, data were Fourier-transformed and the 405 nM excitation signal was subtracted from the 465 nM excitation signal to account for movement artifacts and photobleaching ($\Delta F/F$). The subtracted signal was smoothed using a custom fifth order bandpass Butterworth filter (cutoff frequencies: 0.05 and 2.25 Hz) and returned to the time domain using custom scripts written in MATLAB (version R2022b, Mathworks). To compare changes in fluorescence across recording sessions and across rats in different experimental groups, the processed signal for each recording session was normalized to the session's average fluorescence and converted to $z$-scores. The normalized signal was then aligned to events of interest (*Figure 1B*). Custom MATLAB scripts are available in the public repository, Github (https://github.com/maxineloh/loh-elife-2024, copy archived at *Loh, 2025*; GNU General Public License v3.0).

## Intraoral infusion protocols

Before each session, intraoral catheters were flushed with distilled water to ensure patency. All training and experimental sessions took place in standard chambers (ENV-009A-CT, Med Associates Inc) or custom-made cylindrical chambers for behavioral reactivity recordings. At the start of a session, a fluid line from a syringe containing a test solution was gravity-fed into a two-way solenoid valve (The Lee Company) and the fluid line exiting the solenoid was connected to the intraoral catheter. A TTL from Med Associates software and hardware triggered the opening and closing of the solenoid valve. Rats were habituated to receive 30 brief (5 s, 200 µl) intraoral infusions of water repeated at varying intertrial intervals (35–55 s) for 2 days. Intraoral delivery ensured that rats tasted the stimulus but were not obligated to consume it. This session structure was used throughout all experimental conditions detailed below.

### Primary taste

Rats received intraoral infusions of either 0.3 M sucrose or 0.001 M quinine and received the other solution the following day with solution order counterbalanced across rats (*Figure 1E*).

### Single-pairing CTA

After habituation, all rats underwent a 3-day training cycle. The first day was CD, where rats received intraoral infusions of 0.3 M sucrose and were randomly assigned to either the Paired or Unpaired group. Immediately after the CD session, rats were injected intraperitoneally (i.p.; 20 ml/kg) with either 0.15 M LiCl for Paired rats or 0.15 M NaCl for Unpaired rats. On the second day, rats did not receive intraoral infusions. Instead, they were injected with the counterbalanced treatment and returned to the home cage. On the third day, rats were left undisturbed in their home cage. This 3-day CTA training cycle was followed by the TD, in which rats received a session of intraoral infusions of sucrose with no subsequent injections.

### Single-pairing CTA with extinction

Treatment of Paired and Unpaired animals was the same as Single-pairing CTA above except that TD served as the first (E1) of five extinction sessions (E1–E5). For each extinction day, a session of intraoral infusions was administered as on TD. Extinction sessions were administered on consecutive days.

### Delayed-test CTA

The same procedure as the Single-pairing CTA paradigm was used. However, TD was not performed until 7 days after CD, to match the interval from CD to E5 from the Single-pairing CTA with Extinction condition.

### Repeated-pairing CTA with extinction

Paired and Unpaired animals underwent three conditioning cycles (C1–C3) of the procedure described in Single-pairing CTA. Following the conclusion of the third cycle, extinctions sessions were administered for 8 consecutive days (E1–E8).

### Two-Bottle Preference test

After each Extinction session, rats were returned to the home cage and given 2 hr access to two sipper bottles, one containing 0.3 M sucrose and the other water. Bottle locations were switched after 1 hr to avoid a side bias. Change in bottle weight from before to after 2 hr access was used to calculate sucrose preference as sucrose consumed (g)/total fluid consumed (g).

### Behavioral recordings

Behavior in all sessions was video captured at 10 fps for later analysis. In the Primary taste and Single-pairing CTA conditions, rats were recorded in a cylindrical chamber with a clear floor using an adjacent video camera (Teledyne FLIR, Model: BFS-U3-13Y3M-C) that captured behavioral reactivity via the reflection from an angled mirror positioned under the chamber (below view, *Grill and Norgren, 1978a*). In the studies of Single- and Repeated-pairing CTA with Extinction and Delayed-test CTA, rats were recorded in standard Med Associates chambers with a camera (Wo-We Webcam 720P) positioned above the chamber (above view). Video was monitored using recording software (below view: Teledyne FLIR, Spinnaker SDK; above view: Synapse, Tucker Davis Technologies) and stored for off-line analysis. A custom Med Associates program, linked to the camera's operation, sent TTLs to time stamp the onset and offset of the video recordings on the fiber photometry software (Synapse, Tucker Davis Technologies) to later sync video to photometry recordings.

Behavioral reactivity was calculated as the change in movement from the 5-s pre-infusion period to the 5-s intraoral infusion. Positional coordinates for body parts of the rat were obtained using the open-source deep-neural network toolbox, DeepLabCut (*Mathis et al., 2018*). A custom DeepLabCut model was developed and implemented to track the nose, forepaws, and chamber legs (for scale) in the below view recordings. For above view recordings, a different DeepLabCut model was developed to track the ears and chamber corners. Custom MATLAB codes were employed to calculate the midpoint between the ears (reported as head position) and track distance moved during specified time points, using the coordinates from stationary landmarks (i.e., chamber legs and corners) to normalize distances from pixels to metric units. Any coordinates with model likelihood estimates below 95% were excluded (≤9% of timepoints) and replaced with the linear interpolation of coordinates at neighboring high-confidence timepoints.

### Immunohistochemistry

Following completion of experiments, rats were deeply anesthetized with isoflurane and transcardially perfused with 0.9% NaCl followed by 10% buffered formalin solution (HT501320, Sigma-Aldrich). Brains were removed and stored in formalin and switched to 20% sucrose the following morning. All brains were sectioned at 40 μm on a freezing stage microtome (SM2010R, Leica Biosystems). Sections were collected and processed to fluorescently tag GFP (as an indicator of GCaMP6f or GRAB_DA2h expression) and/or TH via immunohistochemistry. Primary antibodies were incubated at 4°C (washes and other steps at room temperature). Tissues were washed with 1× potassium phosphate-buffered saline (KPBS) six times for 10 min, permeabilized in 0.3% Triton X-100 for 30 min and blocked in 2% normal donkey serum for 30 min. Sections were incubated in rabbit anti-TH (AB152, Sigma-Aldrich) and/or chicken anti-GFP (AB13907, Abcam) antibodies overnight (~18 hr). After six 10-min 1× KPBS washes, secondary antibody (Cy3 conjugated donkey anti-rabbit and AF488 conjugated donkey anti-chicken; Jackson Immunoresearch) was applied and sections were incubated for 2 hr at room temperature followed by a single wash. Sections were then mounted onto glass slides, air dried, and coverslipped with Fluoroshield with DAPI (F6057, Sigma-Aldrich). Data from rats with GFP expression and fiber placements within the borders of the VTA or NAc (*Paxinos and Watson, 2007*) were included in analyses.

### Data analyses

To quantify results from in vivo fiber photometry experiments, the mean *z*-score during the 5-s infusion period was measured on each trial and averaged across trials for each session. One- or two-way

repeated measures ANOVA or paired *t*-tests were used for statistical comparisons. When group main effects were found with two or more treatments, Tukey's (for one-way ANOVAs) and Šidák's (for two-way ANOVAs) post hoc tests were employed. Linear regression was used to calculate p-values, $r^2$ goodness-of-fit, 95% confidence bands of the best-fit line, and linear equations for average dopamine response versus average behavioral reactivity or sucrose preference score. A p-value of <0.05 was used to determine statistical significance. These statistical analyses were performed using GraphPad Prism 10.0 Software (GraphPad Software Inc).

Finally, we performed ROC analysis (*Green and Swets, 1966*; *Cone et al., 2015*) to quantify how discriminable neural responses were to different test conditions via R Studio using the pROC package (package version 1.18.5, RStudio Team (2020), code available on GitHub *Robin et al., 2011*). For each trial, we measured the mean *z*-score during the 5-s intraoral infusion period. The analysis shows the ability of an ideal observer to classify a neural response as driven by sucrose/Unpaired infusion (compared with quinine/Paired). We plotted the rate of hits as a function of false alarm rate across a range of thresholds determined as the mean of the difference between consecutive values of observed *z*-scores. We then computed the area under the ROC curve ($AUC_{ROC}$) and used a threshold of $AUC_{ROC}$ >0.7 to indicate that an ideal observer could reliably discriminate between conditions. All data are available as figure source data files and linked to associated figures.

## Acknowledgements

We gratefully acknowledge the help of Drs. Ted Hsu and Vaibhav Konanur for technical support in fiber photometry training and analysis.

## Additional information

### Funding

| Funder | Grant reference number | Author |
|---|---|---|
| National Institute of Diabetes and Digestive and Kidney Diseases | T32DK128782 | Maxine K Loh |
| National Institute on Drug Abuse | R01DA025634 | Jamie D Roitman Mitchell F Roitman |

The funders had no role in study design, data collection and interpretation, or the decision to submit the work for publication.

### Author contributions

Maxine K Loh, Data curation, Formal analysis, Supervision, Funding acquisition, Writing – original draft, Writing – review and editing; Samantha J Hurh, Paula Bazzino, Data curation; Rachel M Donka, Data curation, Formal analysis, Visualization; Alexandra T Keinath, Formal analysis, Visualization, Methodology; Jamie D Roitman, Formal analysis, Supervision, Writing – review and editing; Mitchell F Roitman, Conceptualization, Formal analysis, Supervision, Funding acquisition, Methodology, Project administration, Writing – review and editing

### Author ORCIDs

Maxine K Loh ⬩ https://orcid.org/0000-0002-2933-2768
Mitchell F Roitman ⬩ https://orcid.org/0000-0003-3973-635X

### Ethics

All studies were conducted in accordance with the National Institutes for Health Guide for the Care and Use of Laboratory Animals and approved by the Animal Care Committee at the University of Illinois Chicago (#22-119).

Reviewer #1 (Public review): https://doi.org/10.7554/eLife.103260.2.sa1
Reviewer #2 (Public review): https://doi.org/10.7554/eLife.103260.2.sa2

Reviewer #3 (Public review): https://doi.org/10.7554/eLife.103260.2.sa3
Author response https://doi.org/10.7554/eLife.103260.2.sa4

## Additional files

### Supplementary files
MDAR checklist

### Data availability
All data generated and code written for analyses are made publicly available. Custom MATLAB scripts are available in the public repository, Github (https://github.com/maxineloh/loh-elife-2024 copy archived at *Loh, 2025*). All data are available as figure source data files and linked to associated figures.

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
