## [Editor Report · eLife Assessment]

This study utilizes an elegant approach to examine valence encoding of the mesolimbic dopamine system. The findings are **valuable**, demonstrating differential responses of dopamine to the same taste stimulus according to its valence (i.e., appetitive or aversive) and in alignment with distinct behavioral responses. The evidence supporting the claims is **convincing**, resulting from a well-controlled experimental design with minimal confounds and thorough reporting of the data.

---

## [Referee Report · Reviewer #1 (Public review)]

Summary:

Loh and colleagues investigate valence encoding in the mesolimbic dopamine system. Using an elegant approach, they show that sucrose, which normally evokes strong dopamine neuron activity and release in the nucleus accumbens, is made aversive via conditioned taste aversion, the same sucrose stimulus later evokes much less dopamine neuron activity and release. Thus, dopamine activity can dynamically track the changing valence of an unconditioned stimulus. These results are important for helping clarify valence and value related questions that are the matter of ongoing debate regarding dopamine functions in the field.

Strengths:

This is an elegant way to ask this question, the within subject's design and the continuity of the stimulus is a strong way to remove a lot of the common confounds that make it difficult to interpret valence-related questions. I think these are valuable studies that help tie up questions in the field while also setting up a number of interesting future directions. There are number of control experiments and tweaks to the design that help eliminate a number of competing hypotheses regarding the results. The data are clearly presented and contextualized.

Weaknesses for consideration:

The focus on one relatively understudied region of the rat striatum for dopamine recordings could potentially limit generalization of the findings. While this can be determined in future studies, the implications should be further discussed in the current manuscript.

---

## [Referee Report · Reviewer #2 (Public review)]

Summary:

Koh et al. report an interesting manuscript studying dopamine binding in the lateral accumbens shell of rats across the course of conditioned taste aversion. The question being asked here is how does the dopamine system respond to aversion? The authors take advantage of unique properties of taste aversion learning (notably, within-subjects remapping of valence to the same physical stimulus) to address this.

They combine a well controlled behavioural design (including key, unpaired controls) with fibre photometry of dopamine binding via GrabDA and of dopamine neuron activity by gCaMP, careful analyses of behaviour (e.g., head movements; home cage ingestion), the authors show that, (1) conditioned taste aversion of sucrose suppresses the activity of VTA dopamine neurons and lateral shell dopamine binding to subsequent presentations of the sucrose tastant; (2) this pattern of activity was similar to the innately aversive tastant quinine; (3) dopamine responses were negatively correlated with behavioural (inferred taste reactivity) reactivity; and (4) dopamine responses tracked the contingency of between sucrose and illness because these responses recovered across extinction of the conditioned taste aversion.

Strengths:

There are important strengths here. The use of a well-controlled design, the measurement of both dopamine binding and VTA dopamine neuron activity, the inclusion of an extinction manipulation; and the thorough reporting of the data. I was not especially surprised by these results, but these data are a potentially important piece of the dopamine puzzle (e.g., as the authors note, salience-based argument struggles to explain these data).

Weaknesses for consideration:

(1) The focus here is on the lateral shell. This is a poorly investigated region in the context of the questions being asked here. Indeed, I suspect many readers might expect a focus on the medial shell. So, I think this focus is important. But, I think it does warrant greater attention in both the introduction and discussion. We do know from past work that there can be extensive compartmentalisation of dopamine responses to appetitive and aversive events and many of the inconsistent findings in the literature can be reconciled by careful examination of where dopamine is assessed. I do think readers would benefit from acknowledgement this - for example it is entirely reasonable to suppose that the findings here may be specific to the lateral shell.

(2) Relatedly, I think readers would benefit from an explicit rationale for studying the lateral shell as well as consideration of this in the discussion. We know that there are anatomical (PMID: 17574681), functional (PMID: 10357457), and cellular (PMID: 7906426) differences between the lateral shell and the rest of the ventral striatum. Critically, we know that profiles of dopamine binding during ingestive behaviours there can be highly dissimilar to the rest of ventral striatum (PMID: 32669355). I do think these points are worth considering.

(3) I found the data to be very thoughtfully analysed. But in places I was somewhat unsure:

(a) Please indicate clearly in the text when photometry data show averages across trials versus when they show averages across animals.

(b) I did struggle with the correlation analyses, for two reasons.

(i) First, the key finding here is that the dopamine response to intraoral sucrose is suppressed by taste aversion. So, this will significantly restrict the range of dopamine transients, making interpretation of the correlations difficult.

(ii) Second, the authors report correlations by combining data across groups/conditions. I understand why the authors have done this, but it does risk obscuring differences between the groups. So, my question is: what happens to this trend when the correlations are computed separately for each group? I suspect other readers will share the same question. I think reporting these separate correlations would be very helpful for the field - regardless of the outcome.

(4) Figure 1A is not as helpful as it might be. I do think readers would expect a more precise reporting of GCaMP expression in TH+ and TH- neurons. I also note that many of the nuances in terms of compartmentalisation of dopamine signalling discussed above apply to ventral tegmental area dopamine neurons (e.g. medial v lateral) and this is worth acknowledging when interpreting.

---

## [Referee Report · Reviewer #3 (Public review)]

Summary:

This study helps to clarify the mixed literature on dopamine responses to aversive stimuli. While it is well accepted that dopamine in the ventral striatum increases in response to various rewarding and appetitive stimuli, aversive stimuli have been shown to evoke phasic increases or decreasing depending on the exact aversive stimuli, behavioral paradigm, and/or dopamine recording method and location examined. Here the authors use a well-designed set of experiments to show differential responses to an appetitive primary reward (sucrose) that later becomes a conditioned aversive stimulus (sucrose previously paired with lithium chloride in a conditioned taste aversion paradigm). The results are interesting and add valuable data to the question of how the mesolimbic dopamine system encodes aversive stimuli, however, the conclusions are strongly stated given that the current data do not necessarily align with prior conflicting data in terms of recording location, and it is not clear exactly how to interpret the generally biphasic dopamine response to the CTA-sucrose which also evolves over exposures within a single session.

Strengths:

• The authors nicely demonstrate that their two aversive stimuli examined, quinine and sucrose following CTA, evoked aversive facial expressions and paw movements that differed from those following rewarding sucrose to support that the stimuli experienced by the rats differ in valence.

• Examined dopamine responses to the exact same sensory stimuli conditioned to have opposing valences, avoiding standard confounds of appetitive and aversive stimuli being sensed by different sensory modalities (i.e., sweet taste vs. electric shock).

• The authors examined multiple measurements of dopamine activity - cell body calcium (GCaMP6f) in midbrain and release in NAc (Grab-DA2h), which is useful as the prior mixed literature on aversive dopamine responses comes from a variety of recording methods.

• Correlations between sucrose preference and dopamine signals demonstrate behavioral relevance of the differential dopamine signals.

• The delayed testing experiment in Figure 7 nicely controls for the effect of time to demonstrate that the "rewarding" dopamine response to sucrose only recovers after multiple extinction sucrose exposures to extinguish the CTA.

Weaknesses for consideration:

• Regional differences in dopamine signaling to aversive stimuli are mentioned in the introduction and discussion. For instance, the idea that dopamine encodes salience is strongly argued against in the discussion, but the paper cited as arguing for that (Kutlu et al. 2021) is recording from the medial core in mice. Given other papers cited in the text about the regional differences in dopamine signaling in the NAc and from different populations of dopamine neurons in midbrain, it's important to mention this distinction wrt to salience signaling. Relatedly, the text says that the lateral NAc shell was targeted for accumbens recordings, but the histology figure looks like the majority of fibers were in the anterior lateral core of NAc. For the current paper to be a convincing last word on the issue, it would be extremely helpful to have similar recordings done in other parts of the NAc to do a more thorough comparison against other studies.

• Dopamine release in the NAc never dips below baseline for the conditioned sucrose. Is it possible to really consider this as a signal for valence per se, as opposed to it being a weaker response relative to the original sucrose response?

• Related to this, the main measure of the dopamine signal here, "mean z-score," obscures the temporal dynamics of the aversive dopamine response across a trial. This measure is used to claim that sucrose after CTA is "suppressing" dopamine neuron activity and release, which is true relative to the positive valence sucrose response. However, both GRAB-DA and cell-body GCaMP measurements show clear increases after onset of sucrose infusion before dipping back to baseline or slightly below in the average of all example experiments displayed. One could point to these data to argue either that aversive stimuli cause phasic increases in dopamine (due to the initial increase) or decreases (due to the delayed dip below baseline) depending on the measurement window. Some discussion of the dynamics of the response and how it relates to the prior literature would be useful.

- Would this delayed below-baseline dip be visible with a shorter infusion time?

- Does the max of the increase or the dip of the decrease better correlate with the behavioral measures of aversion (orofacial, paw movements) or sucrose preference than "mean z-score" measure used here?

- The authors argue strongly in the discussion against the idea that dopamine is encoding "salience." Could this initial peak (also seen in the first few trials of quinine delivery, fig 1c color plot) be a "salience" response?

• Related to this, the color plots showing individual trials show a reduction in the increases to positive valence sucrose across conditioning day trials and a flip from infusion-onset increase to delayed increases across test day trials. This evolution across days makes it appear that the last few conditioning day trials would be impossible to discriminate from the first few test day trials in the CTA-paired. Presumably, from strength of CTA as a paradigm, the sucrose is already aversive to the animals at the first trial of test day. Why do the authors think the response evolves across this session?

• Given that most of the work is using a conditioned aversive stimulus, the comparison to a primary aversive tastant quinine is useful. However, the authors saw basically no dopamine response to a primary aversive tastant quinine (measured only with GRAB-DA) and saw less noticeable decreases following CTA for NAc recordings with GRAB-DA2h than with cell body GCaMP. Given that they are using the high-affinity version of the GRAB sensor, this calls into question whether this is a true difference in release vs. soma activity or issue of high affinity release sensor making decreases in dopamine levels more difficult to observe.

---

## [Author Response]

**Reviewer #1 (Public review):**
Summary:Loh and colleagues investigate valence encoding in the mesolimbic dopamine system. Using an elegant approach, they show that sucrose, which normally evokes strong dopamine neuron activity and release in the nucleus accumbens, is made aversive via conditioned taste aversion, the same sucrose stimulus later evokes much less dopamine neuron activity and release. Thus, dopamine activity can dynamically track the changing valence of an unconditioned stimulus. These results are important for helping clarify valence and value related questions that are the matter of ongoing debate regarding dopamine functions in the field.

Strengths:

This is an elegant way to ask this question, the within subject's design and the continuity of the stimulus is a strong way to remove a lot of the common confounds that make it difficult to interpret valence-related questions. I think these are valuable studies that help tie up questions in the field while also setting up a number of interesting future directions. There are number of control experiments and tweaks to the design that help eliminate a number of competing hypotheses regarding the results. The data are clearly presented and contextualized.

Weaknesses for consideration:The focus on one relatively understudied region of the rat striatum for dopamine recordings could potentially limit generalization of the findings. While this can be determined in future studies, the implications should be further discussed in the current manuscript.

We agree that the manuscript would benefit from providing a stronger rationale for our recording sites and acknowledging the potential for regional differences in dopamine signaling. We have made the following additions to the manuscript:

Added to the Discussion: “Recordings were targeted to the lateral VTA and the corresponding approximate terminal site in the NAc lateral shell (Lammel et al., 2008). Subregional differences in dopamine activity likely contribute to mixed findings on dopamine and affect. For example, dopamine in the NAc lateral shell differentially encodes cues predictive of rewarding sucrose and aversive footshock, which is distinct from NAc medial shell dopamine responses (de Jong et al., 2019). Our findings are similar to prior work from our group targeting recordings to the NAc dorsomedial shell (Hsu et al., 2020; McCutcheon et al., 2012; Roitman et al., 2008): there, intraoral sucrose increased NAc dopamine release while the response in the same rats to quinine was significantly lower.”

**Reviewer #2 (Public review):**
Summary:Koh et al. report an interesting manuscript studying dopamine binding in the lateral accumbens shell of rats across the course of conditioned taste aversion. The question being asked here is how does the dopamine system respond to aversion? The authors take advantage of unique properties of taste aversion learning (notably, within-subjects remapping of valence to the same physical stimulus) to address this.

They combine a well controlled behavioural design (including key, unpaired controls) with fibre photometry of dopamine binding via GrabDA and of dopamine neuron activity by gCaMP, careful analyses of behaviour (e.g., head movements; home cage ingestion), the authors show that, (1) conditioned taste aversion of sucrose suppresses the activity of VTA dopamine neurons and lateral shell dopamine binding to subsequent presentations of the sucrose tastant; (2) this pattern of activity was similar to the innately aversive tastant quinine; (3) dopamine responses were negatively correlated with behavioural (inferred taste reactivity) reactivity; and (4) dopamine responses tracked the contingency of between sucrose and illness because these responses recovered across extinction of the conditioned taste aversion.

Strengths:There are important strengths here. The use of a well-controlled design, the measurement of both dopamine binding and VTA dopamine neuron activity, the inclusion of an extinction manipulation; and the thorough reporting of the data. I was not especially surprised by these results, but these data are a potentially important piece of the dopamine puzzle (e.g., as the authors note, salience-based argument struggles to explain these data).Weaknesses for consideration:(1) The focus here is on the lateral shell. This is a poorly investigated region in the context of the questions being asked here. Indeed, I suspect many readers might expect a focus on the medial shell. So, I think this focus is important. But, I think it does warrant greater attention in both the introduction and discussion. We do know from past work that there can be extensive compartmentalisation of dopamine responses to appetitive and aversive events and many of the inconsistent findings in the literature can be reconciled by careful examination of where dopamine is assessed. I do think readers would benefit from acknowledgement this - for example it is entirely reasonable to suppose that the findings here may be specific to the lateral shell.

As with our response to Reviewer 1, we agree that we should provide further rationale for focusing our recordings on the lateral shell and acknowledge potential differences in dopamine dynamics across NAc subregions. In addition to the changes in the Discussion detailed in our response to Reviewer 1, we have made the following additions to the Introduction:

Added to the Introduction: “NAc lateral shell dopamine differentially encodes cues predictive of rewarding (i.e., sipper spout with sucrose) and aversive stimuli (i.e., footshock), which is distinct from other subregions (de Jong et al., 2019). It is important to note that other regions of the NAc may serve as hedonic hotspots e.g. dorsomedial shell; or may more closely align with the signaling of salience (e.g. ventromedial shell; (Yuan et al., 2021)).”

(2) Relatedly, I think readers would benefit from an explicit rationale for studying the lateral shell as well as consideration of this in the discussion. We know that there are anatomical (PMID: 17574681), functional (PMID: 10357457), and cellular (PMID: 7906426) differences between the lateral shell and the rest of the ventral striatum. Critically, we know that profiles of dopamine binding during ingestive behaviours there can be highly dissimilar to the rest of ventral striatum (PMID: 32669355). I do think these points are worth considering.

There are several reasons why dopamine dynamics were recorded in the NAc lateral shell:

(1) Dopamine neurons in more medial aspects of the VTA preferentially target the NAc medial shell and core whereas dopamine neurons in the lateral VTA – our target for VTA DA recordings – project to the lateral shell of the NAc (Lammel et al., 2008). Thus, our goal was to sample NAc release dynamics in areas that receive projections from our cell body recording sites.

(2) Cues predictive of reward availability (i.e., sipper spout with sucrose) and aversive stimuli (i.e., footshock) are differentially encoded by NAc lateral shell dopamine, which is distinct from NAc ventromedial shell dopamine responses (de Jong et al., 2019). These findings suggest a role for NAc lateral shell dopamine in the encoding of a stimulus’s valence, which made the subregion an area of interest for further examination.

(3) With respect to the medial NAc shell specifically, extensive literature had already shown it to be a ‘hedonic hotspot’ (Morales and Berridge, 2020; Yuan et al., 2021) whereas the ventral portion is more mixed with respect to valence (Yuan et al., 2021). We had previously shown that intraoral infusions of primary taste stimuli of opposing valence (i.e., sucrose and quinine) evoke differential responses in dopamine release within the NAc dorsomedial shell (Roitman et al., 2008). We more recently replicated differential dopamine responses from dopamine cell bodies in the lateral VTA (Hsu et al., 2020) and thus endeavored to the possibility of changing dopamine responses in the lateral VTA to the same stimulus as its valence changes. As a result of these choices, measuring dopamine release in the lateral shell was a logical choice. The field would greatly benefit from continued future work surveying the entirety of the VTA DA projection terminus.

We have included these points of justification in the Introduction and Discussion sections.

(3) I found the data to be very thoughtfully analysed. But in places I was somewhat unsure:(a) Please indicate clearly in the text when photometry data show averages across trials versus when they show averages across animals.

We have now explicitly indicated in the figure legends of Figures 1, 3, 5, 7, and 8:

(1) In heat maps, each row represents the averaged (across rats) response on that trial.

(2) Traces below heat maps represent the response to infusion averaged first across trials for each rat and then across all rats.

(3) Insets represent the average z-score across the infusion period averaged first across all trials for each rat and then across all rats.

(b) I did struggle with the correlation analyses, for two reasons.(i) First, the key finding here is that the dopamine response to intraoral sucrose is suppressed by taste aversion. So, this will significantly restrict the range of dopamine transients, making interpretation of the correlations difficult.

The overall hypothesis is that the dopamine response would correlate with the valence of a taste stimulus – even and especially when the stimulus remained constant but its valence changed. We inferred valence from the behavioral reactivity to the stimulus – reasoning that an appetitive taste will evoke minimal movement of the nose and paws (presumably because the animals are primarily engaging in small mouth movements associated with ingestion as shown by the seminal work of Grill and Norgren (1978) and the many studies published by the K.C. Berridge group) whereas an aversive taste will evoke significantly more movement as the rats engage in rejection responses (e.g. forelimb flails, chin rubs, etc.). When we conducted our regression analyses we endeavored to be as transparent as possible and labeled each symbol based on group (Unpaired vs Paired) and day (Conditioning vs Test). Both behavioral reactivity and dopamine responses change – but only for the Paired rats across days. In this sense, we believe the interpretation is clear. However, the Reviewer raises an important criticism that there would essentially be a floor effect with dopamine responses. We believe this is mitigated by data acquired across extinction and especially in Figure 9B. Here, the observations that dopamine responses fall to near zero but return to pre-conditioning levels in the Paired group with strong correlation between dopamine and behavioral reactivity throughout would hopefully partially allay the Reviewer’s concerns. See Part ii below for further support.

(ii) Second, the authors report correlations by combining data across groups/conditions. I understand why the authors have done this, but it does risk obscuring differences between the groups. So, my question is: what happens to this trend when the correlations are computed separately for each group? I suspect other readers will share the same question. I think reporting these separate correlations would be very helpful for the field -

regardless of the outcome.

To address this concern, we performed separate regression analyses for Paired and Unpaired rats and provide the table below to detail results where data were combined across groups or separated. Expectedly, all analyses in Paired rats indicated a significant inverse relationship between dopamine and behavioral reactivity. Afterall, it is only in this group where behavioral reactivity to the taste stimulus changes as function of conditioning. Perhaps even more striking is that in almost all comparisons, even when restricting the regression analysis to Unpaired rats, we still observed a significant inverse relationship between dopamine and behavioral reactivity in most experiments. We have outlined the separated correlations below (asterisks denote slopes significantly different from 0; * p<0.05; ** p<0.01; *** p<0.005; **** p<0.001):

**Author response table 1. sa4table1:** 

Fig.	Statistic	Combined Conditions	Unpaired	Paired
4C	R^(2)	0.26	0.10	0.29
	P value	0.0009 (****)	0.2058	0.0123 (*)
	Best Fit Equation	Y=-0.025^(**)X+0.35	Y=-0.017**X+0.51	Y=-0.024^(***)X+0.26
4D	R^(2)	0.22	0.013	0.31
	P value	0.0026 (***)	0.65	0.0093 (**)
	Best Fit Equation	Y=-0.021^(**)X+0.33	Y=-0.0047^(**)X+0.58	Y=-0.025^(**)X+0.23
4G	R^(2)	0.48	0.53	0.55
	P value	0.0002(****)	0.016^(**)	0.0025 (***)
	Best Fit Equation	Y=-0.057^(**)X-0.36	Y=-0.053^(**)X-0.097	Y=-0.057^(***)X-0.52
4H	R^(2)	0.51	0.53	0.54
	P value	<0.0001 (****)	0.017 (*)	{: 0.0026^(******)
	Best Fit Equation	Y=-0.044^(**)X-0.53	Y=-0.040***X-0.25	Y=-0.044^(***)X-0.65
6B	R^(2)	0.38	0.048	0.46
	P value	<0.0001 (****)	0.21	<0.0001 (****)
	Best Fit Equation	Y=0.33^(***)X+0.61	Y=0.037**X+0.86	Y=0.46^(**)X+0.53
9B	R^(2)	0.43	8.4e-005	0.33
	P value	<0.0001 (****)	0.95	<0.0001 (****)
	Best Fit Equation	Y=-0.045^(***)X+0.0036	Y=0.00063^(**)X+0.71	Y=-0.033^(***)X-0.11
9C	R^(2)	0.59	0.15	0.36
	P value	< 0.0001 **** ^("a ")	0.0034 (***)	< 0.0001 (****)
	Best Fit Equation	Y=0.48**X+0.53	Y=0.088^(**)X+0.86	Y=0.44^(**)X+0.48

(4) Figure 1A is not as helpful as it might be. I do think readers would expect a more precise reporting of GCaMP expression in TH+ and TH- neurons. I also note that many of the nuances in terms of compartmentalisation of dopamine signalling discussed above apply to ventral tegmental area dopamine neurons (e.g. medial v lateral) and this is worth acknowledging when interpreting t

Others have reported (Choi et al., 2020) and quantified (Hsu et al., 2020) GCaMP6f expression in TH+ neurons. While we didn’t report these quantifications, our observations were very much in line with previous quantifications from our laboratory (Hsu et al. 2020).

We agree that we should elaborate on VTA subregional differences and have answered this response above (See responses to Reviewer 1 Weakness #1 and Reviewer 2 Weakness #2).

**Reviewer #3 (Public review):**
Summary:This study helps to clarify the mixed literature on dopamine responses to aversive stimuli. While it is well accepted that dopamine in the ventral striatum increases in response to various rewarding and appetitive stimuli, aversive stimuli have been shown to evoke phasic increases or decreasing depending on the exact aversive stimuli, behavioral paradigm, and/or dopamine recording method and location examined. Here the authors use a well-designed set of experiments to show differential responses to an appetitive primary reward (sucrose) that later becomes a conditioned aversive stimulus (sucrose previously paired with lithium chloride in a conditioned taste aversion paradigm). The results are interesting and add valuable data to the question of how the mesolimbic dopamine system encodes aversive stimuli, however, the conclusions are strongly stated given that the current data do not necessarily align with prior conflicting data in terms of recording location, and it is not clear exactly how to interpret the generally biphasic dopamine response to the CTA-sucrose which also evolves over exposures within a single session.Strengths:• The authors nicely demonstrate that their two aversive stimuli examined, quinine and sucrose following CTA, evoked aversive facial expressions and paw movements that differed from those following rewarding sucrose to support that the stimuli experienced by the rats differ in valence.• Examined dopamine responses to the exact same sensory stimuli conditioned to have opposing valences, avoiding standard confounds of appetitive and aversive stimuli being sensed by different sensory modalities (i.e., sweet taste vs. electric shock)• The authors examined multiple measurements of dopamine activity - cell body calcium (GCaMP6f) in midbrain and release in NAc (Grab-DA2h), which is useful as the prior mixed literature on aversive dopamine responses comes from a variety of recording methods.• Correlations between sucrose preference and dopamine signals demonstrate behavioral relevance of the differential dopamine signals.• The delayed testing experiment in Figure 7 nicely controls for the effect of time to demonstrate that the "rewarding" dopamine response to sucrose only recovers after multiple extinction sucrose exposures to extinguish the CTA.Weaknesses for consideration:(1) Regional differences in dopamine signaling to aversive stimuli are mentioned in the introduction and discussion. For instance, the idea that dopamine encodes salience is strongly argued against in the discussion, but the paper cited as arguing for that (Kutlu et al. 2021) is recording from the medial core in mice. Given other papers cited in the text about the regional differences in dopamine signaling in the NAc and from different populations of dopamine neurons in midbrain, it's important to mention this distinction wrt to salience signaling. Relatedly, the text says that the lateral NAc shell was targeted for accumbens recordings, but the histology figure looks like the majority of fibers were in the anterior lateral core of NAc. For the current paper to be a convincing last word on the issue, it would be extremely helpful to have similar recordings done in other parts of the NAc to do a more thorough comparison against other studies.

As the Reviewer notes, NAc dopamine recordings were aimed at the lateral NAc shell. It is possible that some dopamine neurons lying within the anterior lateral core were recorded. Fiber photometry and the size of the fiber optics cannot definitively identify the precise location and number of dopamine neurons from which we recorded. Still, recording sites did not systematically differ between groups. Further, the within-subjects design helps to mitigate any potential biases for one subregion over another. The results presented in the manuscript strongly support a valence code. It is difficult to be the ‘last word’ on this topic and we suspect debate will continue. We used taste stimuli for appetitive and aversive stimuli – whereas many in the field will continue to use other noxious stimuli (e.g. foot shock) that likely recruit different circuits en route to the VTA. And there may very well be a different regional profile for dopamine signaling with different noxious stimuli. Moreover, we used intraoral infusion to avoid confounds of stimulus avoidance and competing motivations (e.g. food or fluid deprivation). We believe that this is one of the most important and unique features of our report. Recent work supports a role for phasic increases in dopamine in avoidance of noxious stimuli (Jung et al., 2024) and it will be critical for the field to reflect on the differences between avoidance and aversion. Moreover, in ongoing studies we aspire to fully survey dopamine signaling in conditioned taste aversion across the medial-lateral and dorsal-ventral axes of the VTA and NAc.

(2) Dopamine release in the NAc never dips below baseline for the conditioned sucrose. Is it possible to really consider this as a signal for valence per se, as opposed to it being a weaker response relative to the original sucrose response?

Indeed, NAc dopamine release to intraoral quinine nor aversive sucrose doesn’t dip below baseline but rather dopamine binding doesn’t change from pre-infusion baseline levels. It should be noted that VTA dopamine cell body activity does indeed dip below baseline in response to aversive sucrose. Moreover, using fast-scan cyclic voltammetry, we showed that dopamine release dips below baseline in the NAc dorsomedial shell in response to intraoral quinine (Roitman et al., 2008). The differences across recording sites may reflect regional differences but they may also reflect differences in recording approaches. GrabDA2h, used here, has relatively slow kinetics that may obscure dips below baseline (see response Weakness# 8 below).

(3) Related to this, the main measure of the dopamine signal here, "mean z-score," obscures the temporal dynamics of the aversive dopamine response across a trial. This measure is used to claim that sucrose after CTA is "suppressing" dopamine neuron activity and release, which is true relative to the positive valence sucrose response. However, both GRAB-DA and cell-body GCaMP measurements show clear increases after onset of sucrose infusion before dipping back to baseline or slightly below in the average of all example experiments displayed. One could point to these data to argue either that aversive stimuli cause phasic increases in dopamine (due to the initial increase) or decreases (due to the delayed dip below baseline) depending on the measurement window. Some discussion of the dynamics of the response and how it relates to the prior literature would be useful.

We have used mean z-score to do much of our quantitative analyses but the Reviewer raises the intriguing possibility that we are masking an initial increase in dopamine release and VTA DA activity evoked by aversive taste by doing so. We included the heat maps in the manuscript to be as transparent as possible about the time course of dopamine responses – both within a trial and across trials. The Reviewer’s point prompted us to reflect further on the heat maps and recognize that trials early in the session often showed a brief increase in dopamine for aversive sucrose but this response dissipated (NAc dopamine release) or flipped (VTA DA cell body activity) over trials. We now quantitatively characterize this feature by looking at the timecourse of dopamine responses in each third of the trials (1-10, 11-20, 21-30; see Author response images 1,2 and 3). As we infer the valence of the stimulus from nose and paw movements (behavioral reactivity), it is especially striking that we a similar timecourse for changes in behavior. Collectively, the data may reflect an updating process that is relatively slow and requires experience of the stimulus in a new (aversive) state – that is, a model-free process. While our experiments were not designed to test the updating of dopamine responses and discern their participation in model-based versus model-free learning processes – another debate in the dopamine field (Cone et al., 2016; Deserno et al., 2021)– the data reflect a model-free process. This is further supported in the experiment involving multiple conditioning sessions, where dopamine ‘dips’ are observed in trials 1-10 on Conditioning Day 3 and Extinction Day 1 when the new value of sucrose has been established. Finally, the relatively slow updating of the value of sucrose is reflected in older literature using a continuous intraoral infusion. Using this approach, rats began rejecting the saccharin infusion only after ~2min rather than immediately (Schafe et al., 1998; Schafe and Bernstein, 1996; Wilkins and Bernstein, 2006).

**Author response image 1. sa4fig1:** 

**Author response image 2. sa4fig2:** 

**Author response image 3. sa4fig3:** 

(4) Would this delayed below-baseline dip be visible with a shorter infusion time?

While our experiments did not explore this parameter, it would be interesting to parametrically vary infusion duration times and examine differences in dopamine responses. However, we believe the most parsimonious explanation is that the ‘dip’ in VTA cell body activity develops as a function of the slow updating of the value of sucrose reflective of a model-free process. We recognize that this is mere speculation.

(5) Does the max of the increase or the dip of the decrease better correlate with the behavioral measures of aversion (orofacial, paw movements) or sucrose preference than "mean z-score" measure used here?

It seems plausible that finding the most extreme value from baseline could better correlate to behavioral measures. Time courses to max increase and max decrease are different. Moreover, with appetitive sucrose, there are often multiple transients that occur throughout a single intraoral infusion. Coupled with a noisy time course for individual components of behavioral reactivity, we determined that averaging data across the whole infusion period (i.e. mean z-score) was the most objective way we could analyze the dopamine and behavioral responses to taste stimuli.

(6) The authors argue strongly in the discussion against the idea that dopamine is encoding "salience." Could this initial peak (also seen in the first few trials of quinine delivery, fig 1c color plot) be a "salience" response?

Our response above to the potential for ‘mixed’ dopamine responses to aversive sucrose led to additional analyses that support a slow updating of both behavior and dopamine to the new, aversive value of sucrose. Quinine is innately aversive and thus the Reviewer rightly points out that even here we observe an increase in dopamine release evoked by quinine on the first few trials (as observed in the heat map). We’d like to note, though, that the order of stimulus exposure was counterbalanced across rats. In those rats first receiving a sucrose session, quinine initially caused a modest increase in dopamine release during the first 10 trials (which is more pronounced in the first 2 trials). In the subsequent 2 blocks of 10 trials, no such increase was observed. Interestingly, in rats for which quinine was their first stimulus, we did not see an increase in dopamine release on the first few trials (see Author response image 4). We speculate that the initial sucrose session required the value of intraoral infusions to be updated when quinine was delivered to these rats and that, once more, the updating process may be slow and akin to a model-free process. This analysis, at present, is underpowered but will direct future attention in follow-up work.

**Author response image 4. sa4fig4:** 

(7) Related to this, the color plots showing individual trials show a reduction in the increases to positive valence sucrose across conditioning day trials and a flip from infusion-onset increase to delayed increases across test day trials. This evolution across days makes it appear that the last few conditioning day trials would be impossible to discriminate from the first few test day trials in the CTA-paired. Presumably, from strength of CTA as a paradigm, the sucrose is already aversive to the animals at the first trial of test day. Why do the authors think the response evolves across this session?

As the Reviewer noted, Points 3-7 are related. We have speculated that the evolving dopamine response in Paired rats across test day trials reflects a model-free process. Importantly, as in the manuscript, our additional analyses once again show a tight relationship between behavioral reactivity and the dopamine response across the test session trials. It is important to note, though, that these experiments were not designed to test if responses reflect model-free or model-based processes.

(8) Given that most of the work is using a conditioned aversive stimulus, the comparison to a primary aversive tastant quinine is useful. However, the authors saw basically no dopamine response to a primary aversive tastant quinine (measured only with GRAB-DA) and saw less noticeable decreases following CTA for NAc recordings with GRAB-DA2h than with cell body GCaMP. Given that they are using the high-affinity version of the GRAB sensor, this calls into question whether this is a true difference in release vs. soma activity or issue of high affinity release sensor making decreases in dopamine levels more difficult to observe.

We share the same speculation as the Reviewer. Using fast-scan cyclic voltammetry, albeit measuring dopamine concentration in the dorsomedial shell, we observed a clear decrease from baseline with intraoral infusions of quinine (Roitman et al., 2008). Using fiber photometry here, the Reviewer and we note that GRAB_DA2h is a high-affinity (i.e., EC50: 7nM) dopamine sensor with relatively long off-kinetics (i.e., t1/2 decay time: 7300ms) (Labouesse et al., 2020). It may therefore be much more difficult to observe decreases (below baseline) using this sensor. The publication of new dopamine sensors - with lower affinity, faster kinetics, and greater dynamic range (Zhuo et al., 2024) – introduces opportunities for comparison and the greater potential for capturing decreases below baseline. Due to the poorer kinetics associated with GRAB_DA2h, we would not assert that direct comparisons between the GCaMP- and GRAB-based signals observed here represent true differences between somatic and terminal activity.

References

Choi JY, Jang HJ, Ornelas S, Fleming WT, Fürth D, Au J, Bandi A, Engel EA, Witten IB. 2020. A Comparison of Dopaminergic and Cholinergic Populations Reveals Unique Contributions of VTA Dopamine Neurons to Short-Term Memory. *Cell Rep* 33. doi:10.1016/j.celrep.2020.108492

Cone JJ, Fortin SM, McHenry JA, Stuber GD, McCutcheon JE, Roitman MF. 2016. Physiological state gates acquisition and expression of mesolimbic reward prediction signals. *Proc Natl Acad Sci U S A* 113. doi:10.1073/pnas.1519643113

de Jong JW, Afjei SA, Pollak Dorocic I, Peck JR, Liu C, Kim CK, Tian L, Deisseroth K, Lammel S. 2019. A Neural Circuit Mechanism for Encoding Aversive Stimuli in the Mesolimbic Dopamine System. *Neuron* 101. doi:10.1016/j.neuron.2018.11.005

Deserno L, Moran R, Michely J, Lee Y, Dayan P, Dolan RJ. 2021. Dopamine enhances model-free credit assignment through boosting of retrospective model-based inference. *Elife* 10. doi:10.7554/eLife.67778

Hsu TM, Bazzino P, Hurh SJ, Konanur VR, Roitman JD, Roitman MF. 2020. Thirst recruits phasic dopamine signaling through subfornical organ neurons. *Proc Natl Acad Sci U S A* 117:30744–30754. doi:10.1073/PNAS.2009233117/-/DCSUPPLEMENTAL

Jung K, Krüssel S, Yoo S, An M, Burke B, Schappaugh N, Choi Y, Gu Z, Blackshaw S, Costa RM, Kwon HB. 2024. Dopamine-mediated formation of a memory module in the nucleus accumbens for goal-directed navigation. *Nat Neurosci*. doi:10.1038/s41593-024-01770-9

Labouesse MA, Cola RB, Patriarchi T. 2020. GPCR-based dopamine sensors—A detailed guide to inform sensor choice for in vivo imaging. *Int J Mol Sci*. doi:10.3390/ijms21218048

Lammel S, Hetzel A, Häckel O, Jones I, Liss B, Roeper J. 2008. Unique Properties of Mesoprefrontal Neurons within a Dual Mesocorticolimbic Dopamine System. *Neuron* 57. doi:10.1016/j.neuron.2008.01.022

McCutcheon JE, Ebner SR, Loriaux AL, Roitman MF, Tobler PN. 2012. Encoding of aversion by dopamine and the nucleus accumbens. *Front Neurosci* 6. doi:10.3389/fnins.2012.00137

Morales I, Berridge KC. 2020. ‘Liking’ and ‘wanting’ in eating and food reward: Brain mechanisms and clinical implications. *Physiol Behav*. doi:10.1016/j.physbeh.2020.113152

Roitman MF, Wheeler RA, Wightman RM, Carelli RM. 2008. Real-time chemical responses in the nucleus accumbens differentiate rewarding and aversive stimuli. *Nature Neuroscience 2008 11:12* 11:1376–1377. doi:10.1038/nn.2219

Schafe GE, Bernstein IL. 1996. Forebrain contribution to the induction of a brainstem correlate of conditioned taste aversion: I. The amygdala. *Brain Res* 741. doi:10.1016/S0006-8993(96)00906-7

Schafe GE, Thiele TE, Bernstein IL. 1998. Conditioning method dramatically alters the role of amygdala in taste aversion learning. *Learning and Memory* 5. doi:10.1101/lm.5.6.481

Wilkins EE, Bernstein IL. 2006. Conditioning method determines patterns of c-fos expression following novel taste-illness pairing. *Behavioural Brain Research* 169. doi:10.1016/j.bbr.2005.12.006

Yuan L, Dou YN, Sun YG. 2021. Topography of reward and aversion encoding in the mesolimbic dopaminergic system. *Journal of Neuroscience* 39. doi:10.1523/JNEUROSCI.0271-19.2019

Zhuo Y, Luo B, Yi X, Dong H, Miao X, Wan J, Williams JT, Campbell MG, Cai R, Qian T, Li F, Weber SJ, Wang L, Li B, Wei Y, Li G, Wang H, Zheng Y, Zhao Y, Wolf ME, Zhu Y, Watabe-Uchida M, Li Y. 2024. Improved green and red GRAB sensors for monitoring dopaminergic activity in vivo. *Nat Methods* 21. doi:10.1038/s41592-023-02100-w